# Sustained freshening of Arabian Sea High Salinity Water induced by extreme precipitation events

Prasad G. Thoppil 

In the northern Arabian Sea, high salinity levels are primarily sustained by year-round evaporation, driving the convective formation of Arabian Sea High Salinity Water (ASHSW) during the winter monsoon (November–February). Although precipitation has largely been discounted as a critical controlling mechanism for winter convection, recent years have seen a notable increase in extreme cyclones over the Arabian Sea, particularly in post-monsoon cyclones (September–December) since 2014. However, the extent to which these cyclone-induced freshwater inputs disrupt the region's freshwater balance (evaporation – precipitation) and impact ASHSW formation remains unclear. Here, we present observational evidence supported by a suite of model simulation experiments, revealing a significant weakening in ASHSW formation triggered and sustained by extreme tropical cyclones. The addition of freshwater reduces the density of high-salinity water, augmenting stratification and disrupting the convective sinking process, ultimately limiting the depth of convective mixing. These findings underscore the profound implications of extreme cyclone-induced freshwater inputs.

Dominated by evaporation year-round, the northern Arabian Sea sustains high salinity water greater than 36.5 psu. During winter (November–February), the cool and dry air brought by continental northeasterly winds enhances evaporative cooling, which, combined with reduction in solar radiation, leads to a net heat loss from the ocean[1–3]. As a result, the surface water becomes denser through a decrease in sea surface temperature (SST) and an increase in salinity. This denser water then sinks, leading to the convective formation of Arabian Sea High Salinity Water mass (ASHSW[2,3]). Though heat flux is the dominant contributor to the buoyancy flux over most of the region, salt flux dominates during winter in the northern Arabian Sea[4]. This indicates that evaporative cooling plays a primary role in the formation of ASHSW. After its formation in the north, ASHSW spreads southward along the eastern Arabian Sea at the 26-isopycnal surface during the southwest monsoon (June–September) by the clockwise basin-scale monsoon circulation. At the equator, ASHSW, characterized by subsurface salinity maxima at 100 m, is advected eastward into the Bay of Bengal and the eastern equatorial Indian Ocean by the prevailing monsoon circulation[5] and the Wyrtki Jet[6].

The year-to-year variations in ASHSW are primarily influenced by changes in evaporation, which are modulated by dry air brought by northeasterly winds[7]. Additionally, mesoscale eddies in the Gulf of Oman contribute to these variations through lateral advection, though to a lesser extent[8,9]. Precipitation has largely been regarded as a minor factor in winter convection, with evaporative cooling being the dominant buoyancy input. However, extreme precipitation in the northeastern Arabian Sea during the summer monsoon of 2017 has been shown to enhance salinity stratification, subsequently shoaling convective mixing depth by 40 m, and weakening ASHSW formation in the following winter[7]. Despite this, such extreme precipitation events are rare and are unlikely to result in sustained long-term changes in convective mixing and ASHSW formation.

The Arabian Sea experiences a bimodal annual frequency distribution of tropical cyclone activity, peaking in both the pre-monsoon and post-monsoon seasons[10]. While pre-monsoon tropical storms have intensified in recent years, there is still no consensus on the mechanisms driving this intensification[11–13]. Post-monsoon extremely severe tropical cyclones were not observed until 2014 but have become more

Ocean Sciences Division, U.S. Naval Research Laboratory, Stennis Space Center, MS 39529, USA. ✉e-mail: prasad.g.thoppil.civ@us.navy.mil

frequent in recent years[14]. This trend aligns with an increase in anthropogenic forcing, such as SST warming and weakened vertical wind shear[14,15]. Additionally, projections indicate an increase in the frequency of post-monsoon extreme cyclone in the Arabian Sea[14]. This projection corresponds with the observed increase in the number of cyclones over the Arabian Sea between 2019 and 2023 compared to the previous 5 years, 2014 to 2018 (Fig. 1a, b). Moreover, the cyclones tracks have notably shifted towards the northern Arabian Sea, specifically north of 15°N where the ASHSW forms. Between 2019 and 2023, 8 out of 11 cyclones tracked north of 15°N, compared to 3 out of 9 cyclones during 2014–2018. In particular, 2019 was one of the most active cyclone seasons on record and the third costliest, with 4 extremely severe cyclones crossing north of 15°N. Traditionally regarded as a minor factor in winter convection, the increasing precipitation from tropical cyclones in recent years could influence winter convection and the formation of ASHSW. However, the extent to which the recent increase in extreme cyclones disrupts the sinking process and weakens the convective formation of ASHSW remains unknown.

Here we show the observational evidence, supported by a suite of model simulation experiments, demonstrating a substantial weakening in ASHSW formation, initiated, and sustained by extreme cyclone events. Since 2019, successive cyclone-induced precipitation events have contributed to the accumulation of low-salinity water, prolonging its presence in the Arabian Sea. The increasing contribution of precipitation from tropical cyclones strengthens stratification through

salinity freshening, stabilizing the water column and potentially impeding winter convection and ASHSW formation.

## Results

### Prolonged salinity freshening: observations

To assess the impact of tropical cyclone-induced precipitation on salinity, we compared precipitation and salinity between the 2019–2023 and 2014–2018 periods. Satellite-derived estimates indicate a notable increase in precipitation over the Arabian Sea during 2019–2023 compared to 2014–2018 (Fig. 1c). The increased precipitation, in excess of 1 mm d$^{-1}$ extending from 10°N northward, aligns with the increased frequency of cyclones along the eastern Arabian Sea (Fig. 1b). The impact of precipitation on the convective formation of ASHSW is clearly evident in the salinity patterns (Fig. 1d). In the eastern Arabian Sea north of 10°N, significant freshening exceeding 0.1 psu occurred during the period 2019–2023 compared to 2014–2018. This salinity freshening aligns with the precipitation anomalies, suggesting that the salinity changes are likely driven by increasing contributions from the extremely severe cyclones.

The Hovmöller diagrams of salinity anomalies align with precipitation anomalies during 2014–2023 (Fig. 2). Elevated salinity during 2014–2018 corresponds to a deficit in precipitation, while the sustained salinity freshening during 2019–2023 aligns with positive precipitation anomalies (Fig. 2a, b). Negative salinity anomalies of 0.2–0.3 psu emerge with the post-monsoon cyclones in 2019, and freshwater

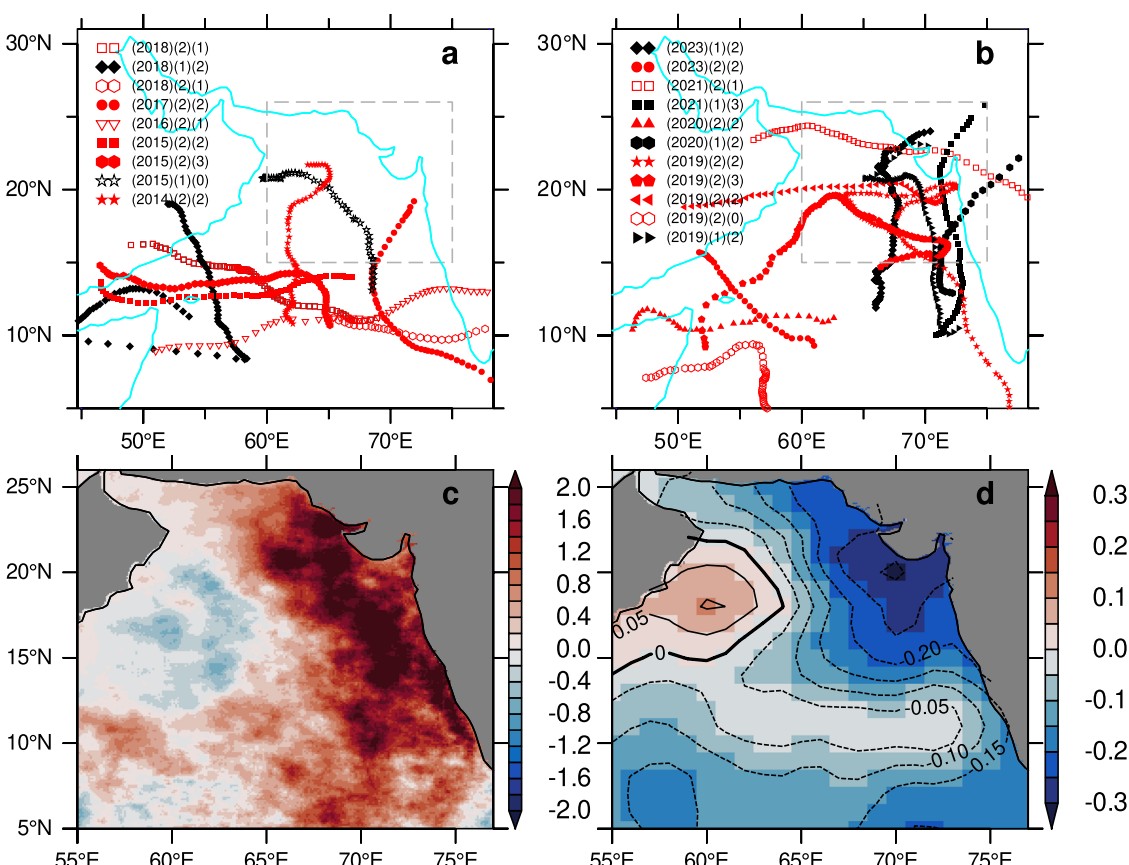

**Fig. 1 | Tropical cyclones and sustained salinity freshening.** Tracks of tropical cyclones during **a** 2014–2018 and **b** 2019–2023, with pre-monsoon (May–June) cyclones marked by black symbols and post-monsoon (September–December) cyclones marked by red symbols. The legend columns represent the year (first column), season (pre-monsoon = 1, post-monsoon = 2) in the second column, and cyclone intensity (0−cyclone, 1−severe, 2−extremely severe, 3−super cyclone) in the last column. Cyclones are categorized by wind speeds, with extreme cyclones defined as those exceeding 46 m s$^{-1}$. During the total period, 20 cyclones

were observed: 6 (30%) during the pre-monsoon and 14 (70%) during the post-monsoon, indicating an increase in post-monsoon cyclones. The number of cyclones north of 15°N increased during 2019–2023. **c**, **d** show the annual mean differences between 2019–2023 and 2014–2018 for **c** rainfall (mm d$^{-1}$) and **d** salinity (psu) in the upper 50 m. Rainfall data are from the Integrated Multi-satellite Retrievals for GPM (IMERG[46]), and salinity data are from the EN4 dataset based on Argo profiles[45]. Positive values represent excess rainfall, while negative salinity values indicate freshening during 2019–2023 compared to 2014–2018.

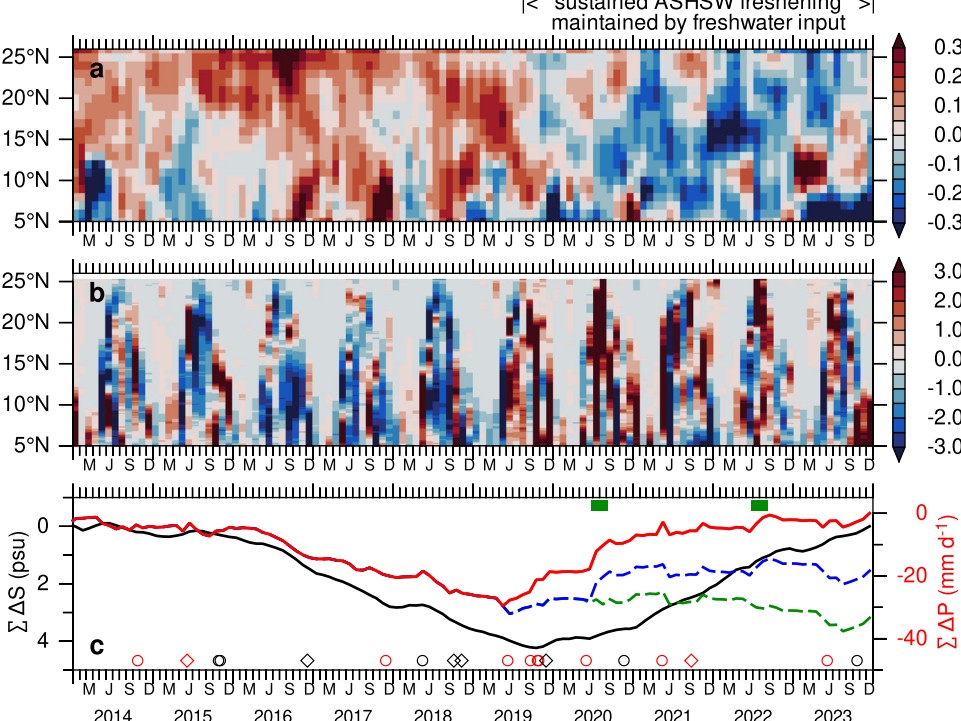

**Fig. 2 | Sustained freshening of Arabian Sea High Salinity Water (ASHSW) driven by freshwater input.** Time-latitude plots of anomalies in **a** salinity in the upper 50 m and **b** precipitation, averaged in the eastern Arabian Sea (60°–75°E). **c** Time-series of cumulative anomalies in salinity (psu) in the upper 50 m (black line) and precipitation (mm d$^{-1}$) (red line) averaged over the region 60°–75°E, 5°–25°N. Cumulative precipitation anomalies excluding all cyclones during 2019–2023 (blue dashed line) and both extreme precipitation events (July–August of 2020 and 2022), combined with cyclones (green dashed line) are also shown (see "Methods").

The two extreme precipitation events in 2020 and 2022 are marked with green boxes. Salinity data are from the monthly EN4 dataset[45], and precipitation data are from the monthly IMERG dataset[46]. Anomalies are calculated relative to the 2000–2023 period by subtracting the long-term monthly climatology. Cyclone occurrences are marked in panel **c**: circles indicate very severe to super cyclones, and diamonds indicate cyclones to severe cyclones. Cyclones penetrating north of 15°N are marked in red.

input increases following the pre-monsoon cyclone, as indicated by the rising cumulative anomalies (Fig. 2c). Although each tropical cyclone during 2019–2023 period is accompanied by an increase in freshwater input (Fig. 2b, c), periods of substantial freshwater gain also occur independently of cyclone activity. In particular, anomalous freshwater input is evident in the northern Arabian Sea during July–August of both 2020 and 2022, marked by rapid increases in cumulative precipitation anomalies (Fig. 2c). These findings highlight the need to distinguish and quantify precipitation contributions from tropical cyclones and other extreme events.

An approach analogous to the framework used to isolate extreme precipitation events originating from tropical cyclones is applied here[16], focusing on the 2019–2023 period. This method involves removing the precipitation contributions within a spatial domain defined by a 750 km radius around the center tracks of each cyclone during this period from the total precipitation (see "Methods"). The time-series of cumulative precipitation anomalies, without the contributions of tropical cyclones, for the eastern Arabian Sea is shown in Fig. 3c (dashed blue line). The difference between total precipitation (solid red line) and precipitation suppressed during tropical cyclone events (dashed blue line) quantifies the contribution of cyclone-induced precipitation to the total (Supplementary Figs. 1 and 2). The cumulative precipitation anomaly excluding cyclones is notably lower than the total precipitation, primarily due to the absence of cyclone-associated freshwater input in 2019.

For non-cyclone periods, extreme precipitation events are identified based on the 95th percentile of precipitation anomalies within the region 60°–75°E and 15°–25°N, a threshold effective in capturing extreme events[17]. Two extreme precipitation events are identified

during July–August of 2020 and 2022, as indicated by the green boxes in Fig. 2c. To isolate the contribution of these extreme events, precipitation is suppressed (set to zero) within the region 60°–75°E and 15°–25°N during July–August of both 2020 and 2022. The time-series of cumulative precipitation anomalies, excluding contributions from both cyclone-induced and non-cyclone periods, is shown in Fig. 2c (green dashed line). The notably lower cumulative precipitation anomalies compared to the total suggest that the extreme precipitation events from tropical cyclones, as well as those in 2020 and 2022, accounted for a significant portion of the freshwater input during the period 2019–2023.

In the remainder of the paper, the term extreme precipitation event is used irrespective of their sources, and salinity freshening specifically refers to the freshening of ASHSW within the upper 50 m, unless stated otherwise. The extreme precipitation is used to encompass both precipitation from tropical cyclones and large-scale synoptic systems, such as depressions or monsoonal lows. While tropical cyclones are transient and highly localized, synoptic systems often persist over broader regions, delivering significant precipitation over time. This effect was evident in the severe flooding in Pakistan in 2022[18] and the record rainfall in India in 2020[19].

The time-series of along-track Argo float observations at three locations in the Arabian Sea ("Methods" and Supplementary Fig. 3), covering both the formation (north) and spreading (south) regions of ASHSW from 2014 to 2023 (Fig. 3) provided additional insights into the salinity freshening observed during 2019–2023. The presence of ASHSW, delineated by salinity greater than 36.5 psu in the upper 100 m, exhibited a relatively consistent high salinity pattern until 2019, with significant freshening observed across all three locations since

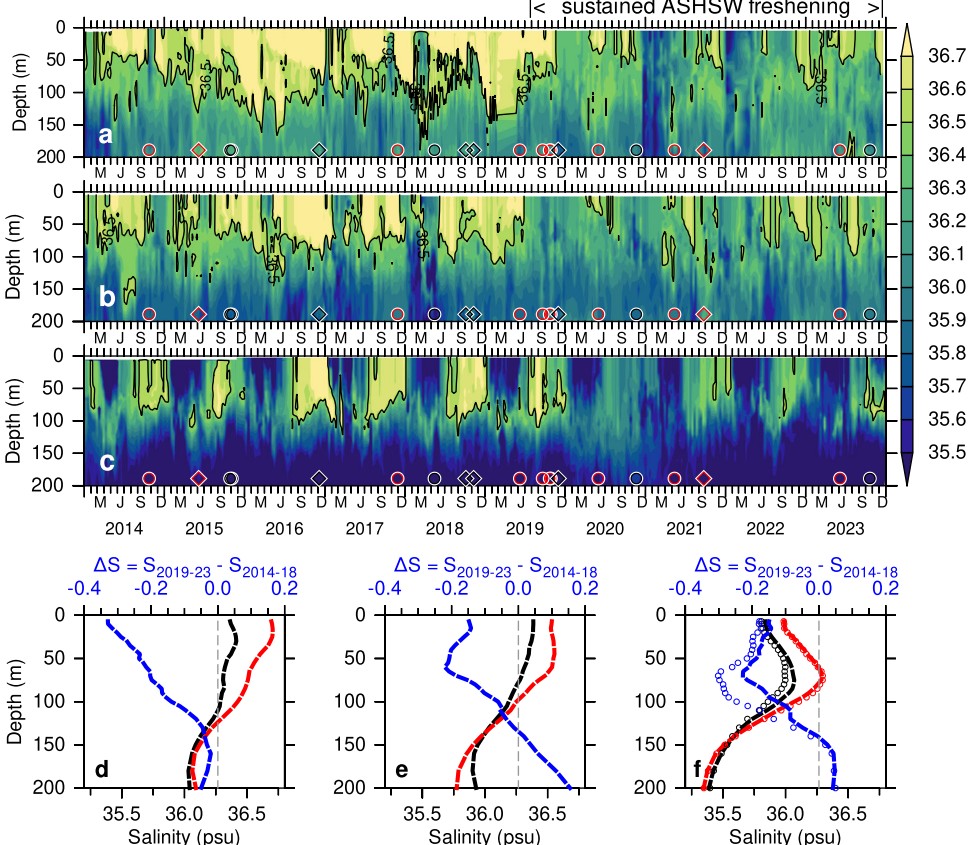

**Fig. 3 | Sustained freshening of Arabian Sea High Salinity Water (ASHSW) revealed by Argo observations.** Along-track Argo float salinity (psu) observations for three regions representing the **a** northern, **b** central, and **c** southern Arabian Sea (refer to Supplementary Fig. 3 for locations) from 2014 to 2023. Cyclone occurrences are marked in each panel: circles indicate very severe to super cyclones, and diamonds indicate cyclones to severe cyclones. Cyclones penetrating north of 15°N are marked in red. **d–f** depict the temporal mean salinity in the upper 200 m for the periods 2014–2018 (red) and 2019–2023 (black), and the differences (blue, $\Delta S = S_{2019\text{-}23} - S_{2014\text{-}18}$) for the three regions shown in (**a–c**). Open circles in **f** represent averages over the periods 2014–2019 and 2020–2023.

then. Following the pre-monsoon cyclone from June 8 to 18, 2019, the salinity decreased by more than 0.5 psu (Fig. 3a, b). This salinity freshening was further sustained by precipitation (Fig. 2) from three post-monsoon, extremely severe cyclones (between September 20 and November 11, 2019), all transversing the northern Arabian Sea north of 15°N. The combined effect of pre- and post-monsoon cyclones in 2019 initiated salinity freshening, and subsequent extreme precipitation events from both cyclones and 2020 and 2022 aided maintain this freshening through 2023.

While extreme precipitation events weaken the convective formation of ASHSW in the north during winter, its presence in the south is influenced by the southward transport of high-salinity water from the north. This transport is facilitated by the prevailing southward current along the eastern Arabian Sea during the summer monsoon. Consequently, the ASHSW exhibited significant freshening of 0.4–0.6 psu in the upper 100 m during the summer of 2020 (Fig. 3c). A pre-monsoon cyclone in June 2020 may have locally contributed to this freshening. Thus, the observations indicate that the northern Arabian Sea was preconditioned with lower salinity due to a pre-monsoon cyclone in June 2019. This stratification was sustained by subsequent post-monsoon cyclones, substantially contributing to the ASHSW freshening in the Arabian Sea.

The eastern Arabian Sea serves as a significant conduit for the southward transport of high-salinity water from the north. During August–October, the ASHSW from the north reaches the southern Arabian Sea, where it forms a weakly stratified layer of high salinity water (>36.5 psu) in the upper 100 m, capped over by relatively fresher water (Fig. 3c), with a subsurface salinity maximum at 75 m. This vertical

salinity stratification promotes vertical mixing (heavier over lighter water) and plays an important role in maintaining SST in the southeastern Arabian Sea[20]. Conversely, in winter (December–February), the westward flowing North Equatorial Current (NEC) carries relatively low-salinity water from the Bay of Bengal, evidenced by salinity <35 psu. During this period, fresher Bay of Bengal water caps over the saltier ASHSW at 100 m, resulting in a strongly stratified upper layer. This layer of freshened water is largely isolated from below by the saltier ASHSW at 100 m. During 2019–2023, as the southern Arabian Sea becomes strongly stratified with fresher ASHSW (Fig. 3c, f), upper-ocean mixing is likely weakened, trapping more heat in the ocean, and contributing to SST warming.

The accumulation of low-salinity water from successive extreme precipitation events each year since 2019 has prolonged the presence of low-salinity water in the Arabian Sea (Figs. 2 and 3). With the exception of 2022, the northern Arabian Sea, north of 15°N, experienced freshwater input from cyclones every year since 2019 (Fig. 1). Although no post-monsoon tropical cyclones reached north of 15°N in 2020 and 2023, freshwater input from pre-monsoon cyclones in May–June 2020 and June 6–16, 2023, contributed to preconditioning the winter, thereby weakening the formation of ASHSW. The pre-monsoon cyclone in 2020, followed by extreme precipitation during July–August, compounded their combined impact on salinity freshening. In 2021, a post-monsoon severe cyclone occurred during September 30–October 4 north of 20°N, and a pre-monsoon super cyclone took place during May 13–19 in the eastern Arabian Sea, further exacerbating the salinity freshening. Although no cyclones occurred in 2022, the extremely high rainfall during July–August,

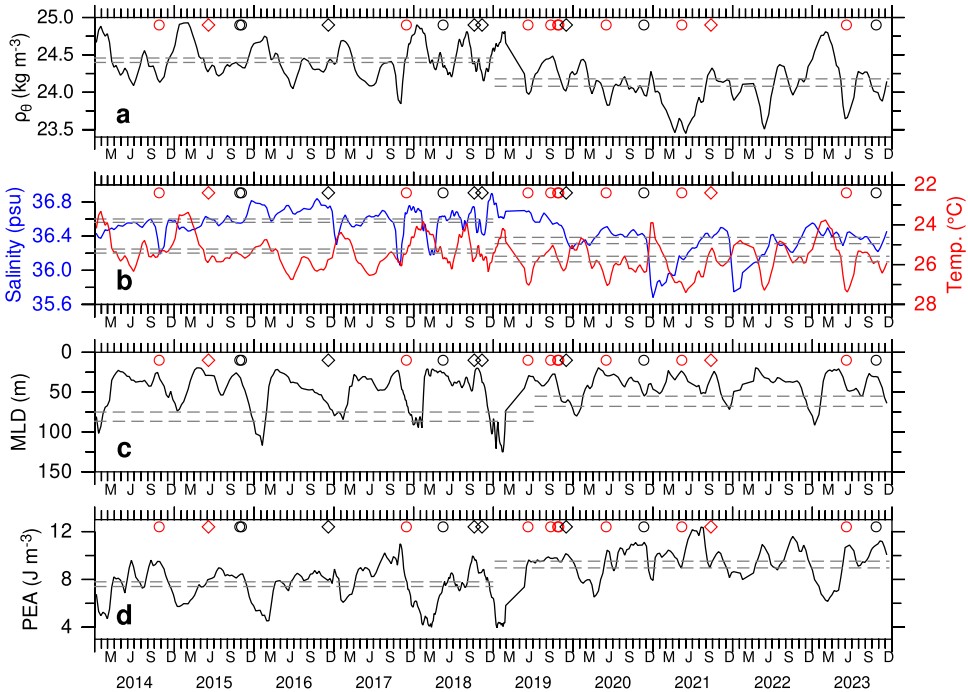

**Fig. 4 | Salinity stratification and the weakening of winter convective mixing.**
**a** Potential density - *1000* (kg m⁻³), **b** salinity (psu, blue line) and temperature (°C, red line) averaged in the upper 100 m, **c** Mixed layer depth (MLD, m), and **d** Potential Energy Anomaly (PEA, J m⁻³) calculated from the Argo profiles in the northern Arabian Sea (Fig. 2a). Gray dashed lines represent the time average values and the error bars over the two 5-year periods for density temperature, salinity, PEA, while MLD is averaged for the December–February period. The difference in means between the two periods are statistically significant at the 95% confidence interval ($p = 0.05$) determined using a standard two-tailed Student's $t$ test. The error bars are calculated as the mean ± 1.96 × standard error of the mean. MLD is calculated from the density increase equivalent to a 0.5 °C temperature decrease from the surface. Cyclone occurrences are marked in each panel: circles indicate very severe to super cyclones, and diamonds indicate cyclones to severe cyclones. Cyclones penetrating north of 15°N are marked in red.

causing severe flooding and displacing over 30 million people in Pakistan[18], contributed to sustaining salinity freshening through 2023. Extreme precipitation during the summer monsoon of 2017 in the northeastern Arabian Sea has been shown to promote stratification, which subsequently weakened the convective formation of ASHSW during the following winter[7]. During 2014–2018, while very severe post-monsoon cyclones in 2014 and 2017 penetrated north of 15°N (Fig. 1) and influenced ASHSW freshening, the formation of ASHSW quickly recovered in subsequent years in the absence of extreme cyclones reaching this region.

The temperature variability in the northern Arabian Sea, unlike salinity, does not show a pronounced difference between the periods 2014–2018 and 2019–2023 (Supplementary Fig. 4). Temperature observations reveal semi-annual variability, with slightly warmer winter temperature observed during 2019–2023. This warming is potentially attributed to weaker vertical mixing, driven by increased salinity stratification. Freshwater input reduces salinity in the upper ocean, creating stable stratification where fresher, less dense water overlays saltier, denser water. This salinity-driven stratification isolates the surface layer from the cooler, denser water below, leading to the formation of a barrier layer[21,22]. This barrier layer suppresses vertical mixing, contributing to the observed winter temperature patterns.

The impact of extreme precipitation on salinity freshening during the periods of 2014–2018 and 2019–2023 is further elucidated in the time-averaged salinity and their differences shown in Fig. 3d–f. Across all three locations, the average salinity during the period 2019–2023 was lower than that of the 2014–2018 period, with a decrease of 0.15–0.3 psu. In the northern Arabian Sea, where ASHSW forms, the largest salinity difference of 0.3 psu occurred at the surface, gradually decreasing down to 120 m. After forming in the northern Arabian Sea, the ASHSW sinks and spreads southward, creating a subsurface maximum. As a result, the most significant salinity freshening in the

southern Arabian Sea was observed at the ASHSW core depth of 75 m (Fig. 3f). Accounting for the time-lag between ASHSW formation in the north and its arrival in the south, the southern region experienced a much larger freshening of 0.3 psu at 75 m, a value comparable to that observed in the north.

The weakening of ASHSW formation in the northern Arabian Sea during 2019–2023 aligned with changes in the strength of stratification and depth of convective mixing (Fig. 4). During 2019–2023, increased freshwater input from extreme precipitation events led to a reduction in upper 100 m density from 24.43 to 24.13 kg m⁻³ (-1% decrease, statistically significant at the 95% confidence interval) compared to 2014–2018 (Fig. 4a). Although the seasonal density variations are primarily controlled by temperature (Fig. 4b), indicating dominance of thermal stratification, the freshwater input can enhance warming by creating barrier layers that inhibit vertical mixing. This is evident in the temperature anomalies, which show periods of warming coinciding with episodes of large freshwater input, particularly in the northern Arabian Sea during 2020–2023 (Supplementary Fig. 5). The resulting reduction in density disrupted the sinking process, led to a shallow winter mixed layer, as observed during 2019–2023 (Fig. 4c). On average, the winter mixed layer shoaled by ~20 m (23.9%, statistically significant at the 90% confidence interval) during 2019–2023 (61.49 m) compared to 2014–2018 (80.79 m). The winter mixing was notably weaker during the winters of 2020–2021 and 2021–2022, with similarly reduced convective mixing apparent in 2014–2015 following a very severe cyclone. The weakening of winter convective mixing is further evident in the mixed layer depth anomalies derived from the EN4 (Supplementary Fig. 6). This pattern is particularly pronounced during February from 2020 to 2023, with the mixed layer consistently shallower by 5–25 m. To quantify stratification strength, the Potential Energy Anomaly (PEA, J m⁻³) was calculated, representing the amount of mechanical energy required to fully mix the water column[23]. A PEA

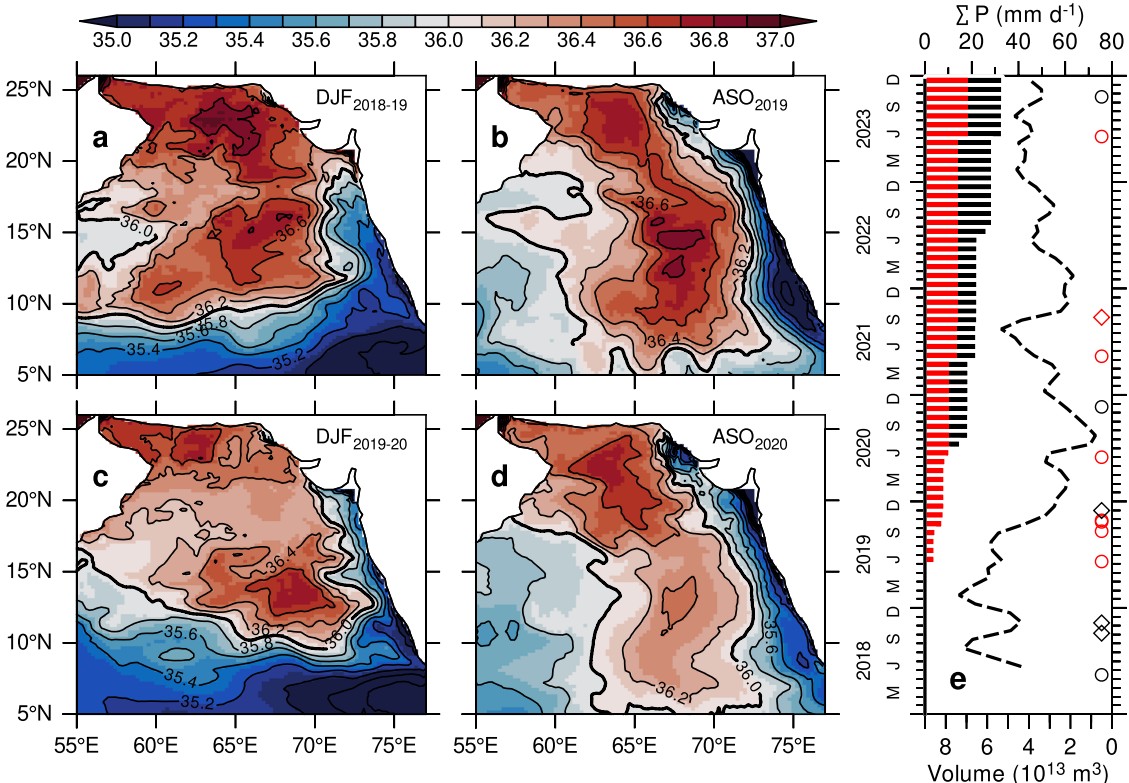

**Fig. 5 | Contrasting precipitation forcing on Arabian Sea High Salinity Water (ASHSW) formation and spreading.** Seasonal mean salinity averaged in the upper 50 m from the control model experiment (CNTL) for **a** winter 2018–2019 (December–February), **b** summer 2019 (August–October), **c** winter 2019–2020, **d** summer 2020. **e** Time-series of volume-integrated salinity (≥36.6 psu) representing ASHSW production for the period 2018–2023 from the CNTL experiment.

**e** includes the cumulative precipitation contributions from cyclones (red histograms), and from the combined influence of cyclones and extreme precipitation events (black histograms), based on IMERG data. Contour interval for salinity is 0.2 psu. Cyclone occurrences are marked in (**e**): circles indicate very severe to super cyclones, and diamonds indicate cyclones to severe cyclones. Cyclones penetrating north of 15°N are marked in red.

value of zero indicates a fully mixed (homogenous) water column and becomes increasingly positive as stratification increases (see "Methods"). PEA decreased during winter due to convective mixing driven by evaporative cooling and increased from spring, reaching its peak in summer (Fig. 4d). While seasonal stratification is evident throughout the period, it is notably stronger during 2019–2023 compared to 2014–2018. Enhanced stratification was evident in each winter and summer since 2019, indicated by larger PEA, coinciding with freshening events driven by extreme precipitation. The time-averaged PEA in the northern Arabian Sea increased by 22% (7.605 to 9.246 J m$^{-3}$) during 2019–2023 compared to 2014–2018. This highlights the role of surface buoyancy forcing from precipitation as a key factor in the development of enhanced stratification.

**Contrasting ASHSW formation: model simulations**
To supplement the spatially and temporally limited Argo observations and confirm the physical processes driving the freshening of ASHSW, we employed results from regional ocean circulation model simulations covering the period 2018–2023 (see "Methods"). Although the model simulations do not span the full 2014–2023 period, they capture the recent salinity freshening events from 2019 to 2023. We specifically examined two contrasting periods: 2018–2019 and 2019–2020. In 2018, although three cyclones occurred, none induced salinity freshening north of 15°N, the region critical for ASHSW formation. In contrast, in 2019, four out of five cyclones penetrated north of 15°N, coinciding with large salinity freshening (Fig. 2). The model reproduced the contrasting salinity patterns observed in Argo data between the 2018–2019 and 2019–2020 (Supplementary Figs. 7 and 8), increasing confidence in the reliability of the model results.

The impact of sustained precipitation from extreme cyclones in 2019 on the formation and spreading of ASHSW is evident in the seasonal mean salinity, averaged in the upper 50 m (Fig. 5). During the winter of 2018–2019 (December–February), the formation of ASHSW is characterized by salinity levels exceeding 36.6 psu in the northern Arabian Sea, north of 18°N (Fig. 5a). The high salinity water (36.6 psu) seen between 10° and 18°N comprises remnants from the preceding summer monsoon. By the late summer monsoon (August–October, Fig. 5b), the southward advection of ASHSW along the eastern Arabian Sea results in salinity levels exceeding 36 psu north of 5°N[2,24,25]. In contrast, the winter of 2019–2020 reveals a significant weakening of ASHSW, with salinity levels exceeding 36.6 psu confined to a small area in the northwestern Arabian Sea and the Gulf of Oman (Fig. 5c). A pronounced reduction in salinity, reaching below 36.2 psu between 15°N and 22°N, can be attributed to three extremely severe cyclones that occurred from September to November 2019 (Fig. 1). The salinity freshening in the northern Arabian Sea, initiated by extreme cyclones, spreads southward along the eastern Arabian Sea during the following summer, demonstrating its persistence (Fig. 5d). Consequently, the salinity in the eastern Arabian Sea decreases by 0.2–0.3 psu compared to the summer of 2019. These findings are consistent with Argo observations (Fig. 3), further strengthening confidence in the model results.

To quantify changes in ASHSW between 2018–2019 and 2019–2020, we estimated the volume-integrated salinity[8] of ≥36.6 psu (Fig. 5e). The ASHSW volume typically peaks during its formation period, with a secondary maximum during the late summer monsoon. However, in 2019, the volume of high-salinity water exhibited a notable decline coinciding with cyclone-induced precipitation events (Fig. 5e).

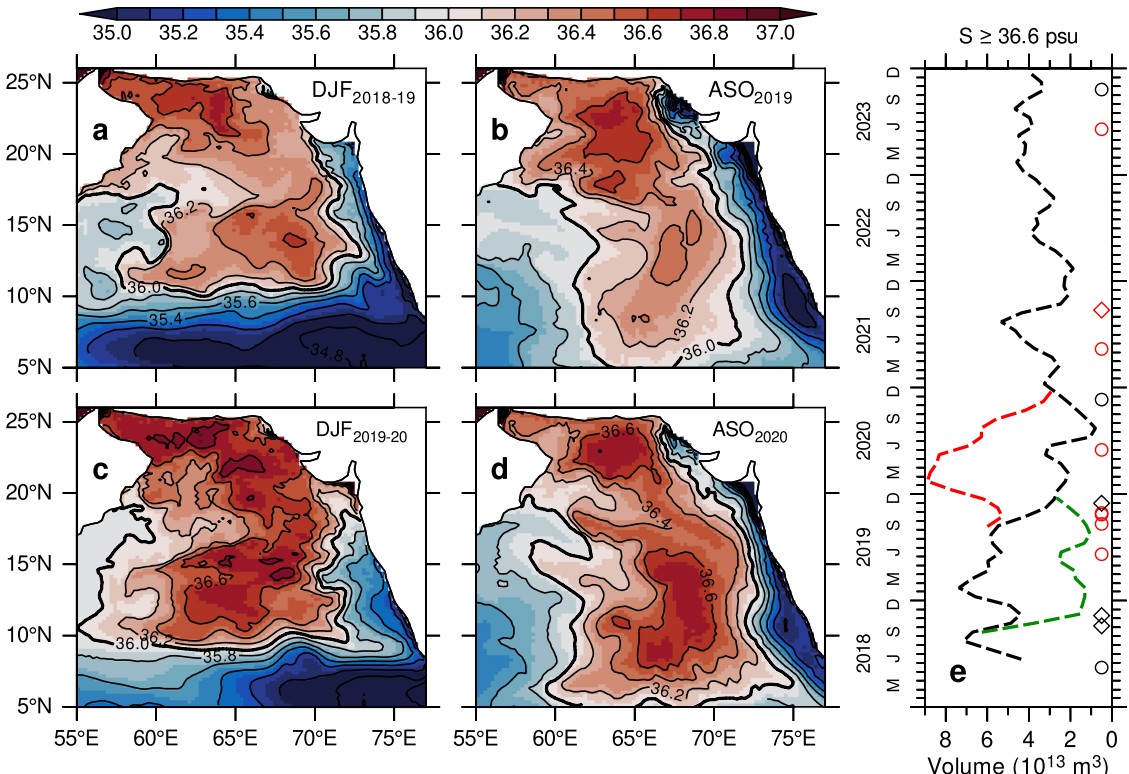

**Fig. 6 | Sensitivity of Arabian Sea High Salinity Water (ASHSW) formation to precipitation: model experiments.** Seasonal mean salinity in the upper 50 m from the perturbation model experiment PPE$_{2018-19}$ for **a** winter 2018–2019 (December–February) and **b** summer 2019 (August–October), and from the PPE$_{2019-20}$ experiment for **c** winter 2019–2020 and **d** summer 2020. **e** Time-series of volume-integrated salinity (≥36.6 psu) representing ASHSW production for the period 2018–2023 from the PPE$_{2018-19}$ (green dashed line), PPE$_{2019-20}$ (red dashed line), and CNTL (black dashed line) experiments. The contour interval for salinity is 0.2 psu. Cyclone occurrences are marked in (**e**): circles indicate very severe to super cyclones, and diamonds indicate cyclones to severe cyclones. Cyclones penetrating north of 15°N are marked in red.

Specifically, the volume of ASHSW decreased steadily from 2019 to the summer of 2020. Between the winters of 2018–2019 and 2019–2020, ASHSW volume decreased by 60%, from 6.3 to 2.5 × 10$^{13}$, and by 77% (from 5.2 to 1.2 × 10$^{13}$) in the subsequent August–October period. By the end of 2023, ASHSW production remained consistently lower (<4 × 10$^{13}$) compared to 2018–2019 winter levels. This decline is consistent with the salinity freshening trends observed in the Argo data (Fig. 3).

**Isolating cyclone-induced precipitation effects on ASHSW freshening: case studies**

Building on the model's accurate representation of the contrasting ASHSW conditions between 2018–2019 and 2019–2020, we conducted a series of experiments to isolate the influence of cyclone-induced precipitation on the salinity freshening by modifying the precipitation forcing (see "Methods"). The first experiment, referred to as PPE$_{2018-19}$, is designed to simulate the cyclone-induced freshening in 2018–2019 by prescribing precipitation and winds from the 2019–2020 period while retaining all other forcings from 2018–2019. This experiment specifically focuses on the post-monsoon cyclones in 2019 and incorporates precipitation from all post-monsoon cyclones in 2019 (including 3 extreme and 1 severe cyclones) as well as two cyclones from 2020. The model is initialized on September 1, 2018 from the control simulation (CNTL) and integrated through December 2019. By modifying only precipitation and winds, this simulation isolates their role in driving salinity changes, enabling attribution of any simulated salinity differences to these factors. The PPE$_{2018-19}$ simulation successfully replicated the pronounced salinity freshening in the northern Arabian Sea during winter and its subsequent southward advection along the eastern Arabian Sea in the following summer (Fig. 6a, b).

These results confirm that the post-monsoon cyclone-induced precipitation in 2019 was the primary driver of salinity freshening in the northern Arabian Sea during winter. Despite retaining 2018–2019 forcing for all parameters except precipitation and winds, the simulated salinity distribution in PPE$_{2018-19}$ closely matched that of the CNTL during 2019–2020 (Figs. 5 and 6). The volume of ASHSW in the PPE$_{2018-19}$ experiment decreased rapidly by 77%, from 6.2 to 1.4 × 10$^{13}$ m$^3$, between September and December 2018, closely mirroring the changes observed in CNTL during 2019 (Fig. 6e).

In a second experiment, termed PPE$_{2019-20}$, we aimed to suppress the cyclone-induced freshening observed during 2019–2020 by prescribing precipitation and winds from the 2018–2019 period, while retaining other forcings from 2019–2020. Notably, neither of the two post-monsoon cyclones from 2018 penetrated north of 15°N (Fig. 1), thereby having no discernible influence on ASHSW formation. The model was initialized on September 1, 2019 from the CNTL simulation, and integrated through December 2020. By replacing the post-monsoon extreme cyclones from 2019 with those from 2018, the salinity freshening failed to replicate in the PPE$_{2019-20}$ experiment. The resulting salinity distribution in PPE$_{2019-20}$ (Fig. 6c, d) closely resembled that of the CNTL simulation for 2018–2019 (Fig. 5a, b). In the northern Arabian Sea, salinity levels exceeding 36.6 psu during winter indicate convective formation of ASHSW (Fig. 6c), while its southward advection along the eastern Arabian Sea leads to a notable increase in ASHSW (36.6 psu) during the following summer (Fig. 6d). In the PPE$_{2019-20}$, simulation, ASHSW volume increased rapidly during the winter (Fig. 6e, red line). Additional sensitivity experiments focusing solely on precipitation perturbations, while excluding winds, yielded similar results (not shown). These model experiments reinforce the conclusion that cyclone-induced precipitation was the primary driver

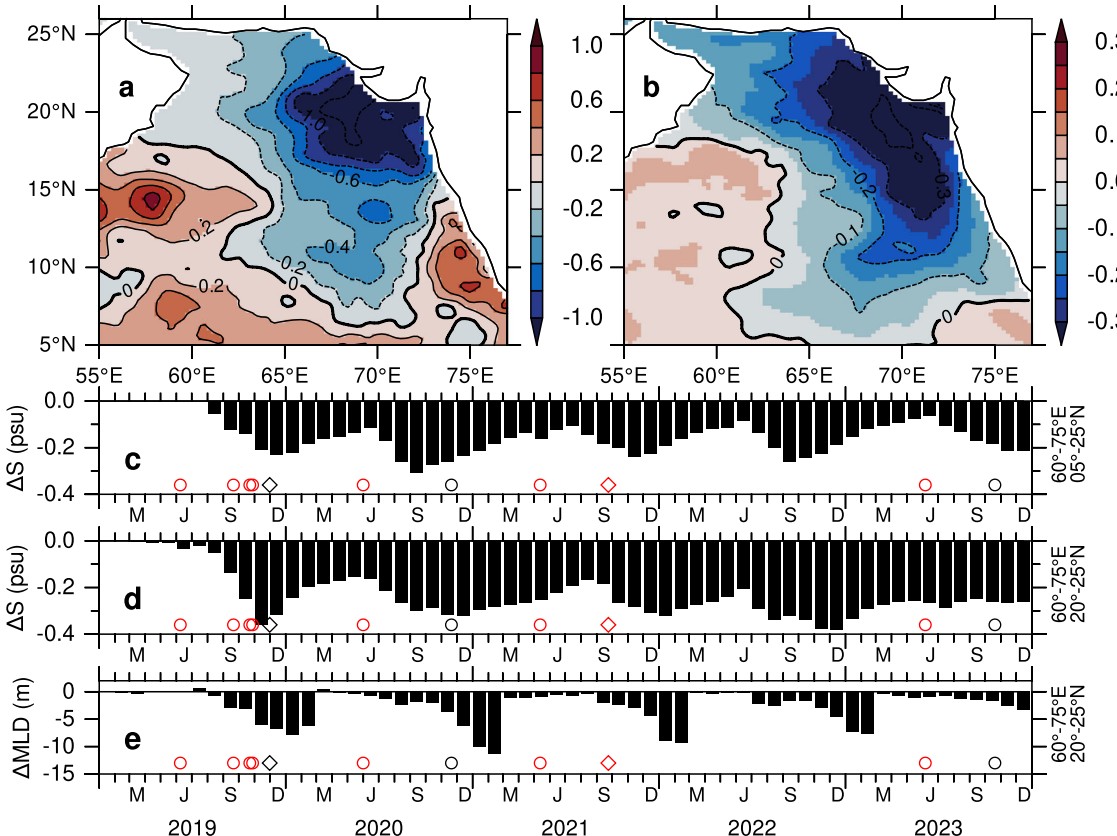

**Fig. 7 | Arabian Sea High Salinity Water (ASHSW) freshening from model perturbation experiment.** Differences between the control (CNTL) and PPE$_{2019\text{-}23}$ experiments (CNTL–PPE$_{2019\text{-}23}$) for the 5-year period spanning 2019–2023 in **a** surface salt flux ($10^{-6}$ psu m s$^{-1}$) and **b** salinity in the upper 50 m. Temporal evolution of salinity differences in the upper 50 m for **c** the eastern Arabian Sea (60°–75°E, 5°–25°N), **d** the northern Arabian Sea (60°–75°E, 20°–25°N), and **e** mixed layer depth (MLD, m) in the northern Arabian Sea (60°–75°E, 20°–25°N). MLD is calculated from the density increase equivalent to a 0.5 °C temperature decrease from the surface. Negative values indicate (**b**) decrease in salinity due to (**a**) excess precipitation over evaporation (**c**, **d**) lower salinity (**e**) shallower MLD in the CNTL compared to PPE$_{2019\text{-}23}$ experiment.

of the observed salinity freshening in the Arabian Sea during the 2019–2020.

## Role of precipitation anomalies in ASHSW freshening (2019–2023)

The above case studies clearly demonstrate that the extreme cyclones of 2019 contributed to upper ocean salinity stratification through increased precipitation. Sustained stratification is achieved only when the buoyancy input from freshwater outcompetes with the vertical mixing driven by winds and evaporative cooling processes. This raises an important question: did the elevated precipitation levels associated with tropical cyclones during 2019–2023 play a crucial role in maintaining stratification and prolonging salinity freshening in the northern Arabian Sea? To address this question, we conducted an additional perturbation experiment, referred to as PPE$_{2019\text{-}23}$ (see "Methods"). In this experiment, the precipitation forcing from 2018, a year characterized by minimal cyclone influence on ASHSW formation, is applied consistently throughout the 2019–2023 simulation period, while all other forcings are maintained according to their respective years. This approach allows us to isolate the impact of precipitation anomalies, primarily from tropical cyclones, on the observed freshening during 2019–2023. The model is initialized in January 2019 from the control simulation and is integrated through December 2023.

Comparisons between PPE$_{2019\text{-}23}$ and the control simulation highlight the contribution of precipitation anomalies to salinity changes during the 2019–2023 period relative to a typical year. Figure 7 illustrates the differences in surface salt flux and salinity between PPE$_{2019\text{-}23}$ and the control simulation. The surface salt flux, driven by

the freshwater flux (evaporation minus precipitation), is negative in the eastern Arabian Sea, indicating a loss of salt due to increased freshwater input (Fig. 7a). The large salt flux of 0.2–1 × 10$^{-6}$ psu m s$^{-1}$ corresponds to a significant reduction in salinity, exceeding 0.1–0.3 psu in the eastern Arabian Sea (Fig. 7b). Since the simulations differ only in precipitation while maintaining nearly identical evaporation rates, the salinity differences are attributed solely to differences in precipitation. Interestingly, the differences in salt flux and salinity between the model experiments (Fig. 7a, b) closely resemble the observed differences in precipitation and salinity between the 2014–2018 and 2019–2023 periods (Fig. 1c, d), further corroborating the critical role of precipitation anomalies in salinity freshening during the latter period.

The temporal evolution of salinity differences further highlights the sustained salinity freshening driven by precipitation anomalies relative to 2018 (Fig. 7c, d). This freshening becomes evident following the post-monsoon cyclones in 2019, with salinity decreasing by 0.1–0.2 psu. With the exception of 2022[18], there is a clear agreement between precipitation from cyclones penetrating north of 15°N and the intensification of salinity freshening. The influence of these cyclones on ASHSW formation and the associated weakening of winter convective mixing in the northern Arabian Sea is illustrated in Fig. 7d, e. A significant reduction in salinity by 0.2–0.3 psu during winter, indicative of weakened ASHSW formation, is evident following the cyclones, with the largest effects in 2019 and 2021, while 2020 also coincides with an extreme precipitation event. The mixed layer depth shoals by 5–10 m each winter, indicating weaker convective mixing due to stronger salinity stratification. This shoaling is most pronounced during the

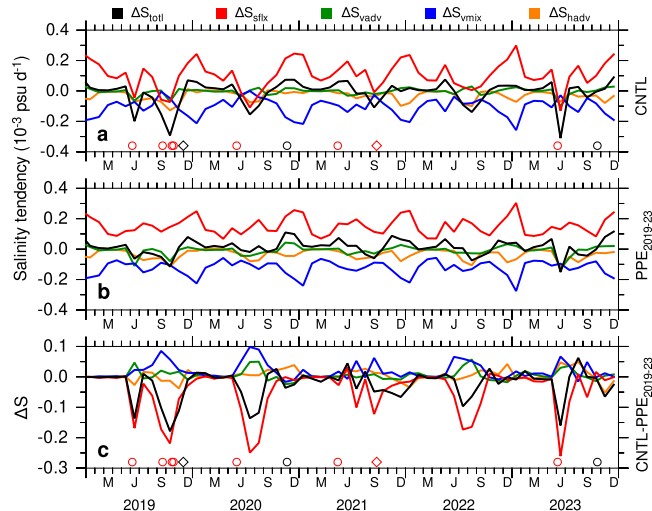

**Fig. 8 | Analysis of salinity-budget terms.** Volume averaged monthly mean salinity budget terms ($10^{-3}$ psu d$^{-1}$) in the northern Arabian Sea (60°–70°E, 15°–25°N, 0–100 m) from the **a** control (CNTL), and **b** PPE$_{2019-23}$ perturbation experiment and **c** differences between CNTL and PPE$_{2019-23}$ ($\Delta S$). The salinity budget terms include tendency term ($\Delta S_{totl}$, black) due to surface freshwater flux ($\Delta S_{sflx}$, red), horizontal advection ($\Delta S_{hadv}$, orange), vertical advection ($\Delta S_{vadv}$, green), and vertical mixing ($\Delta S_{vmix}$, blue). Cyclone occurrences are marked in panels **a** and **c**: circles indicate very severe to super cyclones, and diamonds indicate cyclones to severe cyclones. Cyclones penetrating north of 15°N are marked in red.

active convective mixing period (December–February), when wind-driven mixing is relatively weak. These results suggests that strong stratification, preconditioned by freshwater input, limits convective mixing to shallower depths. Consistent trends in MLD are also observed in Argo float data (Fig. 4b). Thus, the model experiments further confirm that the observed salinity freshening and the associated weakening of convective mixing during 2019–2023 are primarily driven by freshwater input from tropical cyclones and other extreme precipitation events.

## Processes governing ASHSW freshening: salinity budget analysis

To quantify the physical processes controlling salinity freshening during the period 2019–2023, we computed the salinity budget terms (see "Methods") from the control and PPE$_{2019-23}$ simulations, as presented in Fig. 8a. Salinity tendency is governed by contributions from surface freshwater fluxes (evaporation – precipitation + river runoff), vertical mixing, horizontal advection, and vertical advection. Although horizontal mixing is present, it is considered negligibly small compared to the other terms and is therefore not included. Excess evaporation over precipitation removes freshwater from the ocean, resulting in a positive contribution to the salinity budget. Conversely, excess precipitation and river run off add freshwater to the ocean, thereby reducing the salinity of the water column. Due to the vertical gradient of salinity in the northern Arabian Sea, with salinity decreasing with depth (Fig. 3), vertical mixing consistently brings lighter water from below into the mixed layer. The contributions of horizontal and vertical advection to salinity changes are small, as they tend to cancel each other out. Overall, the surface fluxes dominate the tendency term of the salinity budget during the 2019–2023 period, primarily driving increases in salinity. In particular, surface fluxes make a substantial contribution to salinity budget in winter, leading to the formation of ASHSW[3]. The vertical mixing acts to reduce salinity by mixing fresher water from below, partially compensating the salinity increase caused by surface fluxes. The occurrence of extreme cyclones coincides with significant decreases in salinity, driven primarily by surface freshwater fluxes.

The salinity budget terms in the PPE$_{2019-23}$ exhibit similar seasonal changes, with surface fluxes dominating the salinity tendency and the vertical mixing term partially accounting for these differences (Fig. 8b). The differences in salinity budget terms between the control and PPE$_{2019-23}$ experiments elucidate the processes driving the freshening of ASHSW during the 2019–2023 period (Fig. 8c). Each year, the salinity tendency term shows significant decreases, with values ranging from 0.05 to $0.15 \times 10^{-3}$ psu d$^{-1}$, indicating substantial salinity freshening in the CNTL compared to PPE$_{2019-23}$. This freshening is entirely attributed to surface fluxes, predominantly from precipitation, as evaporation and river run off remains nearly unchanged in these simulations. Although differences in vertical mixing tend to increase salinity, the reduction in salinity due to surface fluxes outweighs the increase caused by vertical mixing. Except for 2022, salinity freshening events are closely linked to freshwater input from pre- and post-monsoon cyclones. In particular, precipitation from extreme cyclones in 2019, 2020, and 2023 lead to a significant salinity reduction by more than $0.1 \times 10^{-3}$ psu d$^{-1}$. Notably, 2020 also coincides with an extreme precipitation event. In 2019 and 2021, the combined impact of pre- and post-monsoon cyclones considerably prolonged the duration of freshwater input, playing a critical role in reducing salinity. Note that the extreme pre-monsoon cyclone of 2021 was confined to the region east of 70°E along the west coast of India. As a result, its direct impact is not accounted for in the salinity budget terms averaged over the region (60°–70°E, 15°–25°N).

The salinity budget analysis highlights the significant impact of extreme precipitation events in both 2020 and 2022 on salinity freshening, even in the absence of named cyclones in 2022. The surface freshwater flux from these extreme event is comparable to that of typical tropical cyclones. This similarity arises from transient nature of cyclones, which deliver intense but short-lived rainfall, whereas prolonged low-pressure systems can contribute substantial freshwater input over an extended period. Collectively, precipitation from both cyclones and extreme precipitation events in 2020 and 2022 played a crucial role in sustaining the salinity freshening observed from 2019 through 2023.

## Isolating precipitation sources driving ASHSW freshening

While the perturbation experiment PPE$_{2019-23}$ successfully isolated the contributions of precipitation anomalies relative to the typical year 2018, the separate contributions from tropical cyclones and extreme precipitation events (blue and green lines in Fig. 2c) on ASHSW freshening remain to be fully elucidated. To address this, we conducted three additional perturbation experiments designed to isolate and quantify these contributions (see "Methods"). In the first experiment, referred to as NOTC, precipitation from all tropical cyclones during 2019–2023 is suppressed. To isolate the impact of extreme precipitation events in 2020 and 2022 on salinity freshening, a second experiment, NOPE, is conducted in which precipitation is set to zero during July–August within the region east of 60°E and north of 15°N. Finally, a third experiment, NOPETC, evaluates the combined impact of tropical cyclones and extreme precipitation events by suppressing precipitation from both sources using the same methodology applied in NOTC and NOPE. In all experiments, other forcings are kept consistent with the respective years, and the model is initialized from the control run (CNTL) on January 1, 2019, and integrated forward through 2023.

The comparison between the experiments (NOTC, NOPE, and NOPETC) and the CNTL run underscores the relative contribution of precipitation anomalies from tropical cyclones, extreme precipitation events, and their combined effect on salinity freshening during the 2019–2023 period. Figure 9 illustrates the salinity changes in these experiments relative to the CNTL run. The salinity differences in all experiments exhibit salinity freshening in the eastern Arabian Sea, indicating that salinity in the control run is lower than in these experiments. The salinity difference between the NOPETC and CNTL

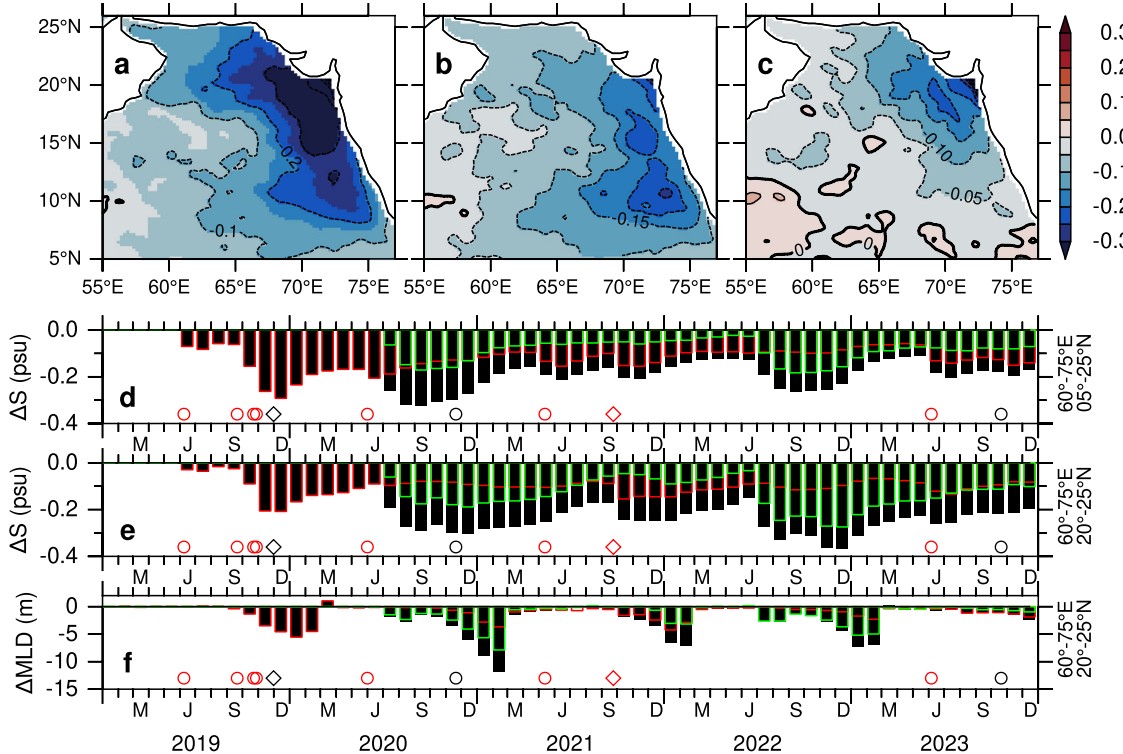

**Fig. 9 | Cyclones and extreme events driving Arabian Sea High Salinity Water (ASHSW) freshening: model experiments.** Salinity differences in the upper 50 m averaged over 2019–2023 between the control run (CNTL) and perturbation experiments: **a** CNTL–NOPETC, **b** CNTL–NOTC, and **c** CNTL–NOPE. Contours are drawn for every **a** 0.1, **b**, **c** 0.05 intervals. Temporal evolution of salinity differences in the upper 50 m for **d** the eastern Arabian Sea (60°–75°E, 5°–25°N), **e** the northern Arabian Sea (60°–75°E, 20°–25°N), and **f** mixed layer depth (MLD, m) in the northern Arabian Sea (60°–75°E, 20°–25°N). Relative contributions from the NOPETC (black), NOTC (red), and NOPE (green) experiments are shown. MLD is calculated from the density increase equivalent to a 0.5 °C temperature decrease from the surface. Negative values represent **a**–**e** salinity freshening and **f** shoaling of the MLD in the CNTL relative to perturbation experiments.

experiments (Fig. 9a) reproduces the overall salinity freshening of 0.2–0.3 psu in the eastern Arabian Sea, with a basin-wide average (60°–75°E, 5°–25°N) freshening of −0.17 psu. The individual contributions of tropical cyclones and extreme precipitation events to salinity freshening are shown in Fig. 9b, c, respectively. Suppressing precipitation from tropical cyclones results in salinity freshening in the region east of 60°E, with freshening increasing to 0.2 psu in the eastern Arabian Sea (70°–75°E). As expected, the areas of enhanced freshening align with tropical cyclone tracks (Fig. 1b). The basin-average salinity freshening is −0.12 psu, accounting for 71% of the total freshening. Similarly, the suppression of precipitation from extreme events in 2020 and 2022 leads to substantial salinity freshening in the eastern Arabian Sea (Fig. 9c). Although precipitation is suppressed in the region north of 15°N, salinity freshening is evident as far as 10°N. This southward extension of freshening occurs as the excess precipitation weakens ASHSW formation in the north and is subsequently advected southward during the following summer. The extreme precipitation events contribute 35% of the total salinity freshening, with a basin-wide average of −0.06 psu.

The temporal evolution of salinity differences between the CNTL and these experiments (Fig. 9d) further illustrates the distinct contributions of salinity freshening caused by tropical cyclones, extreme events, and their combined effects. Notably, salinity freshening becomes evident following the pre- and post-monsoon cyclones of 2019, with a decrease of approximately 0.3 psu during November–December 2019 (Fig. 9d). Freshening in the range of 0.1–0.2 psu is observed after each tropical cyclone crossing north of 15°N. The influence of precipitation events on salinity freshening is particularly pronounced in July 2020 (green histograms), surpassing the impact of tropical cyclones on salinity freshening until December 2020. A similar pattern is evident in 2022.

In the northern Arabian Sea (60°–75°E, 20°–25°N), salinity freshening follows a similar pattern but with some notable differences (Fig. 9e). As expected, significant precipitation from extreme events has a pronounced impact on the formation of ASHSW, with salinity freshening far exceeding that caused by tropical cyclones. Notable salinity freshening is evident following the post-monsoon cyclones in 2019, 2021, as well as a pre-monsoon cyclone in 2023 (red histograms). The salinity-induced stratification contributed to the weakening of winter convection, with stratification driven by extreme events dominating mixed layer depth in 2020 and 2022 (Fig. 9f), whereas cyclone-induced stratification played a more prominent role in reducing the mixed layer depth in 2019 and 2021. The salinity freshening and the corresponding shoaling of the winter mixed layer by 5–10 m (black histograms) are consistent with the results from the PPE$_{2019-23}$ experiment (compare Fig. 9a, d–f and Fig. 7b–e), though with some differences in magnitude. This consistency further reinforces the conclusion that precipitation anomalies from tropical cyclones and extreme events were major contributors to the sustained salinity freshening observed in the eastern Arabian Sea during the 2019–2023 period.

## Discussion

It is anticipated that extreme weather and climate events will become more frequent and severe as a result of climate warming. Recent studies indicate increasing intensity of both pre- and post- monsoon tropical cyclones over the Arabian Sea[12–14]. Specifically, projections suggest a rise in the number of extreme cyclones during the post-monsoon period, along with a discernible northward trend in their tracks[14]. This trend aligns with the recent increase in the number of post-monsoon cyclones, similar to those that occurred in 2019 (Fig. 1). Despite these projections, this study provides valuable insight into the potential impact of cyclone-induced precipitation on salinity,

specifically concerning the formation and seasonal spreading of Arabian Sea High Salinity Water (ASHSW). Through a synthesis of observations and numerical model simulations, we show a sustained freshening of ASHSW caused primarily by precipitation from extreme cyclones over the Arabian Sea. The precipitation reduces density by decreasing surface salinity, which triggers stratification. This strong stratification preconditions the ocean, subsequently weakening the depth of winter convective mixing. Additionally, extreme precipitation events, such as the severe flooding in Pakistan in 2022[18,26,27] in Gujarat in 2017[7,28], and highest August all-India rainfall in the past 44 years recorder in 2020[19], significantly increased freshwater input, further intensifying salinity stratification. While tropical cyclones are transient and localized, synoptic systems such as monsoonal depressions and low-pressure systems often persist over broader regions, delivering substantial precipitation over extended periods. Despite not being linked to named cyclone, precipitation from these systems had a comparable effect on salinity freshening. This study raises important questions about how the increasing intensity and frequency of extreme precipitation events will affect the regional freshwater and salinity budget.

The impact of cyclone-induced precipitation on salinity freshening is typically short-lived, as observed in periods before 2019 (Fig. 3). Two extreme cyclones in 2014 and 2017, and a weak cyclone in 2015 penetrated north of 15°N and influenced salinity freshening (Fig. 1). Although two successive cyclones in 2014 and 2015 contributed to low-salinity ASHSW, the absence of cyclones in 2016 and 2018 provided a period for ASHSW formation to recover. It is worth noting that the cyclone in 2017 veered away from the northern Arabian Sea, tracking northeastward beyond 70°E, potentially diminishing its impact on the formation of ASHSW. In contrast, the accumulated precipitation from four extreme cyclones in 2019 substantially extended the duration of freshwater input, leading to an unprecedented period of prolonged, low salinity conditions (Fig. 3). The accumulation of low-salinity water from successive large precipitation events each year since 2019 has prolonged the presence of low-salinity conditions in the Arabian Sea. This persistent input has hindered the recovery of ASHSW formation, resulting in the sustained freshening of ASHSW from 2019 to 2023.

While precipitation typically affects salinity on shorter timescales, it is expected that salinity levels would quickly recover following a brief precipitation event. However, the cumulative effect of precipitation over the period 2019–2023 far surpassed such transient events, thereby significantly contributing to the maintenance of salinity stratification. The persistence of low salinity water suggests that oceanic dynamical processes played a role in sustaining the freshening over time. Strong winds and ocean currents induce wind-driven mixing and advection, working together to dampen the effects of freshwater-induced salinity changes. Except during cyclone events, wind-driven mixing in the northern Arabian Sea remains relatively weak, thus unable to counteract the stratification induced by freshwater input[7]. Surface currents in the northern Arabian Sea are significantly weaker compared to those in the western Arabian Sea, where the Findlater Jet[29] drives strong currents along the coasts of Somalia, Saudi Arabia, and Oman, along with mesoscale eddies. Consequently, ocean advection plays a smaller role in mitigating the freshwater input, consistent with the minor contribution from advective terms to the salinity tendency term observed in the salinity budget analysis (Fig. 8). The northern Arabian Sea is also a region influenced by the intrusion of high salinity and temperature from the Persian Gulf and Red Sea outflows[30]. Persian Gulf Water typically occurs below 150 m, with its core around 200 m, while Red Sea Water spreads eastward below 500 m. These deeper water masses do not interact with ASHSW, which is largely confined to the upper 100 m, though a portion of ASHSW can be subducted below 150 m during episodes of strong winter convection[8]. The model experiments, which consistently represent these water masses, further support this distinction.

ASHSW is pivotal in governing the salt budget of the Indian Ocean as it mixes with lower salinity water from the Bay of Bengal[31]. In the equatorial Indian Ocean, this water mass forms a subsurface maximum, thereby influencing stratification and mixing processes. An implication is that the strengthening of stratification, resulting from reduced salinity, stabilizes the water column. This stabilization contributes to the warming of SST, particularly pronounced along the eastern Arabian Sea, where the genesis of most tropical cyclones occurs. Increased stratification can also lead to a reduced ocean-atmosphere heat flux and a shallower mixed layer depth, contributing to the genesis of surface intensified marine heat waves[32,33]. In the northern Arabian Sea, the weakening of winter convective mixing due to freshwater-induced stratification may influence primary productivity and habitat availability for ecosystems, potentially altering fisheries catches by shifting species composition[34–37], underscoring the need for further investigation.

An important question arises: what is required for the ASHSW formation cycle to recover? Presently, the low-salinity levels may persist if the northern Arabian Sea, north of 15°N, continues to experience at least one tropical cyclone annually. Projections indicate an increasing number of extreme cyclones[12,14], making this scenario more likely. Furthermore, the increasing frequency of extreme rainfall events may contribute to sustained freshwater input in the region[26,28,38]. This trend may be linked to large-scale climate variability, including the Indian Ocean Dipole (IOD)[39] and Monsoon Intraseasonal Oscillation (MISO)[40]. Notably, MISO-related rainfall has substantially intensified over the northeastern Arabian Sea in the past three decades, driven by enhanced Arabian Sea warming[41]. To a lesser extent, the increasing monsoon precipitation may also contribute to salinity freshening along the west coast of India[42]. On longer climate timescales, recent studies have projected a northward shift of the Intertropical Convergence Zone (ITCZ) from its current baseline position in the Indian Ocean, accompanied by an increase in precipitation[43]. Conversely, under conditions resembling the sustained normal freshwater input observed before 2019, the northern Arabian Sea could see a rebound in ASHSW formation. Regardless of future salinity trends in the Arabian Sea, this study emphasizes the importance of precipitation from tropical cyclones and extreme events as a key factor in salinity stratification and winter convection. Future work should explore the long-term implications of tropical cyclones on primary productivity over decadal timescales.

## Methods
### Potential energy anomaly
The potential energy anomaly[23] (PEA, $\phi$, J m$^{-3}$), is a measure of stratification. It quantifies the amount of mechanical energy per unit volume required to mix the water column vertically homogeneous. The strength of stratification is proportional to $\phi$, with the water column being homogenous when $\phi$ is equal to zero. PEA is defined from the surface to the base of a depth ($h$) is written as:

$$\varphi = \frac{1}{h} \int_{h}^{0} (\hat{\rho} - \rho) g z \, dz \qquad (1)$$

$$\hat{\rho} = \frac{1}{h} \int_{h}^{0} \rho \, dz \qquad (2)$$

where $\rho(z)$ is the potential density profile (kg m$^{-3}$) over a water column of depth $h$ (m), and $\hat{\rho}$ is the vertically averaged potential density (kg m$^{-3}$), $\hat{\rho} - \rho$ is the deviation from the vertically averaged potential density and $g$ (=9.8 m s$^{-2}$) is the acceleration due to gravity. Since our focus is on the upper ocean layer where density stratification occurs, we set the lower limit in the PEA calculation ($h$) to 200 m, which corresponds to the annual maximum of the mixed layer depth. While $\phi$ has traditionally been used in studies of shelf seas, its applicability has

been extended to examining the upper-ocean seasonal stratification in the North Pacific using Argo observations[44].

## ASHSW volume

To quantify the production of Arabian Sea High Salinity Water (ASHSW), we used the following equation for salinity greater than 36.6 psu to compute the volume of ASHSW[8] in m³:

$$V = \iiint (S \ge 36.6) dx dy dz \qquad (3)$$

where, $S$ being the salinity, and $dx$, $dy$, and $dz$ are the zonal, meridional and depth dimensions in meters. In the northern Arabian Sea, salinity greater than 36.5 psu is indicative of newly formed ASHSW (Fig. 3), which is why a threshold of 36.6 psu was chosen. The volume of ASHSW is calculated using the monthly mean salinity from the model experiments.

## Temperature and salinity

We used version 4 of the Met Office Hadley Centre series of datasets (EN4[45]) of global quality-controlled ocean temperature and salinity profiles and monthly objective analyses. The dataset spans from 1900 to present and is based on subsurface ocean temperature and salinity profile data obtained from the World Ocean Database (WOD09), Global Temperature and Salinity Profile Program (GTSPP), Argo, and Arctic Synoptic Basin Wide Oceanography (ASBO) collections. Monthly potential temperature and salinity objective analyses were calculated from the quality-controlled ocean data. These have a regular 1° × 1° horizontal grid and 42 levels in the vertical. This product provides a baseline of salinity for comparing with model results. Salinity anomalies are calculated from 2000–2023 period.

## Argo float observations

In addition to gridded EN4 dataset, we also used individual Argo float observations to identify the periods of anomalous salinity freshening. The accuracies of temperature, salinity and pressure measurements in the Argo profile are ±0.002 °C, ±0.01 psu, and ±2.4 dbar, respectively (https://argo.ucsd.edu/) The Argo data utilized in this study were obtained from argo.ucsd.edu, focusing on three specific locations within the Arabian Sea. These locations were chosen to cover latitudes from 5°N to 25°N in the eastern Arabian Sea. While a single Argo float may not provide complete coverage for the study period spanning from 2014 to 2023, profiles from additional Argo floats within the same grid boxes were stitched together. Stitching was performed based on observation time, resulting in slight discontinuities due to variations in float locations. The locations of these Argo floats separated into three regions are depicted in Supplementary Fig. 3, with each float represented by a different color. For the northern region, four Argo floats with ID's *2902092, 2902132, 2903129, 2902277* are stitched together. In both central and south regions, two Argo floats with ID's *2902263, 2901340* and *2902388, 2901510* are used respectively. Despite this, the stitched data provided seemingly continuous coverage for the entire 2014–2023 period.

## Precipitation

The precipitation observations are obtained from Integrated Multi-satellitE Retrievals for GPM (IMERG) V07 rain rates. IMERG is produced by NASA using a constellation of satellite-based estimates[46]. The IMERG product is created by intercalibrating, merging, and interpolating various precipitation relevant satellite passive microwave (PMW) -based precipitation retrievals, together with PMW-calibrated infrared-based precipitation estimates. Monthly and daily mean rainfall at 0.1° × 0.1° from September 2006 to December 2023 are used. The anomalies are calculated from 2006–2023 period. We also compared precipitation from the National Centers for Environmental

Prediction (NCEP) Climate Forecast System Reanalysis (CFSR[47]). The CFSR precipitation field is almost identical to the IMERG giving confidence in our results (not shown).

## Tropical Cyclone tracks data

Tropical Cyclone (TC) best-track observations were obtained from the International Best Track Archive for Climate Stewardship (IBTrACS, v4r00[48]), which aggregates data from US agencies including National Hurricane Center and Joint Typhoon Warning Center. Our analysis centered on TCs between 2014 and 2023, as 2014 witnessed the first recorded instance of post-monsoon extremely severe cyclonic storms with lifetime maximum winds exceeding 46 m s⁻¹ over the Arabian Sea[14].

## Separating precipitation sources

An approach analogous to isolate precipitation events originating from tropical cyclones is employed here[16], focusing on the 2019–2023 period. This strategy involves removing the precipitation contributions of each cyclone during this period from the total precipitation. Using daily precipitation from IMERG[46] dataset and cyclone track information from IBTrACS[48], precipitation is suppressed (set to zero) during the passage of each storm (±2 days), within a spatial domain defined by a 750 km radius around the cyclone center tracks. Monthly mean precipitation is then constructed from the daily time-series, and anomalies are computed. The cumulative precipitation anomalies excluding the contributions of tropical cyclones for the eastern Arabian Sea, are shown in Fig. 2c (dashed blue line).

For non-cyclone periods, extreme precipitation events are identified based on the 95th percentile of monthly precipitation anomalies within the region 60°–75°E and 15°–25°N, excluding cyclone contributions. This threshold effectively captures extreme events[17]. Two extreme precipitation events are identified during July–August of 2020 and 2022, as indicated by the green boxes in Fig. 2c. To isolate the contribution of these extreme events, precipitation is suppressed (set to zero) within the region 60°–75°E and 15°–25°N during July–August of both years. The time-series of cumulative precipitation anomalies, excluding contributions from both cyclone-induced and non-cyclone extreme precipitation event, is shown in Fig. 2c (green dashed line).

## Model

The study utilizes the Arabian Sea configuration of the Navy Coastal Ocean Model (NCOM[49]), covering the entire Arabian Sea with a horizontal resolution of 25 km. The model domain extends from 45° to 80°E and from 1°S to 32°N, employing 50 vertical levels. The vertical coordinate system employs 34 terrain-following sigma coordinates above 550 m depth and 16 Z or pressure level coordinates below 550 m, with higher resolution near the surface where the surface layer thickness is set to 0.5 m. The Mellor-Yamada level 2 turbulent closure scheme is used for vertical mixing. Forcing for the model includes atmospheric data from the Navy Global Environmental Model (NAVGEM), providing 3-hourly analysis-quality data. This includes surface momentum flux, latent and sensible heat flux, freshwater flux (evaporation – precipitation + river runoff), and solar radiation penetration into the water column. The model uses the monthly mean climatology of river runoff based on the 1996 database[50]. The salinity freshening depicted in the model simulations does not account for the river runoff associated with cyclones and extreme precipitation events. While precipitation from tropical cyclones and extreme events likely increases river runoff, particularly given that some cyclones made landfall along the west of India, their impact on the formation and spreading of ASHSW formation is negligible. Lateral boundary conditions are obtained from the Global Ocean Forecast System (GOFS), a data-assimilative model utilizing the Navy Coupled Ocean Data Assimilation system (NCODA) and the HYbrid Coordinate Ocean Model (HYCOM)[51]. The GOFS is forced with operational NAVGEM atmospheric forcing. The model is initialized on January 1, 2018, from

GOFS analysis and integrated forward through December 31, 2023, providing a comprehensive simulation period for the study.

## Model validation

In general, the model accurately reproduced the formation and seasonal spreading of the Arabian Sea High Salinity Water (ASHSW), consistent with climatological observations[2]. The model results have been validated by comparing them to several Argo float observations during the period 2018–2023. The along-track fields were extracted from the model output using a linear interpolation at the Argo geographical locations. With Argo observations occurring on average every 10 days, a 10-day climatology was constructed using 6 years of data (2018–2023) from two Argo observations in the northern and southern Arabian Sea. The comparisons between the Argo observations and the model demonstrate that the model accurately represents the seasonal salinity cycle in both regions (Supplementary Fig. 7). Although the model shows slightly higher salinity of the ASHSW by -0.2 psu, its vertical extent aligns well with Argo data. The seasonal salinity cycle in the southern Arabian Sea reveals low-salinity water during January–February, which is advected westward from the Bay of Bengal by the North Equatorial Current. This low-salinity water, overlying the saltier ASHSW at 100 m, creates a strong salinity stratification. During the southwest monsoon, ASHSW from the north is advected southward by the prevailing clockwise southwest monsoon circulation[2,25]. Consequently, salinity gradually increases to 36–36.4 psu, with the most significant contribution occurring in September–October. Compared to north, the salinity exhibits a weaker stratification with saltier ASHSW in the upper ocean.

The interannual variation of salinity and temperature in the upper 50 m from the model shows reasonable agreement with the Argo data (Supplementary Fig. 8) with correlation ranging from 0.4 to 0.9. In the northern Arabian Sea, salinity exhibits significant interannual variations, with a notable drop following the pre-monsoon cyclone in 2019. However, in 2022, the model overestimated salinity compared to observations. Salinity ranges between 36 and 36.6 psu, with the highest values occurring in 2018 and early 2019. Conversely, in the southern Arabian Sea, large seasonal salinity variations are observed, ranging from 35 to 36.5 psu. The model salinity aligns more closely with observations in this region than in the northern Arabian Sea. The discrepancies between observed and modeled salinity can be attributed to differences in precipitation and evaporation in the atmospheric forcing, among other model deficiencies. Nonetheless, the significant freshening of ASHSW during the winter of 2019–2020 is well represented in the model, as shown in Fig. 5. We are confident that the model accurately represents the ASHSW freshening induced by tropical cyclones during the period 2018–2023. The temperature exhibits a semiannual cycle, with a primary peak during spring to summer, and a secondary peak during fall to winter, followed by cooling in summer and winter. This variability is driven by net air-sea heat flux and vertical mixing processes.

## Model perturbation experiments

**PPE$_{2019-20}$.** The contrasting ASHSW distribution between the 2018–2019 and 2019–2020 periods are the focus of the first two experiments. We hypothesize that the contrasting ASHSW between these periods resulted from the differences in the precipitation forcing. By prescribing precipitation forcing from a different period is expected to simulate salinity distributions that corresponds to the period of precipitation forcing. The first experiment, referred to as PPE$_{2019-20}$, the model is initialized on September 1, 2018, from the control simulation and integrated through December 2019. The simulation used all but precipitation and winds from the 2018–2019 period. The precipitation and winds are swapped that from the 2019–2020 period. This involved incorporating precipitation from all post-monsoon cyclones in 2019 (including 3 extreme and 1 severe cyclones) and two cyclones in 2020. Despite initial conditions and forcings from 2018–2019 period, the salinity distribution resembled

that from the 2019–2020 control experiment (Fig. 6), confirming that salinity freshening was caused by cyclone induced precipitation.

**PPE$_{2018-19}$.** The second experiment, referred to as PPE$_{2018-19}$, was aimed to suppress the cyclone-induced freshening. To achieve this, the model was initialized on September 1, 2019, from the control run, and integrated through December 2020. We prescribed precipitation and winds from the 2018–2019 period, while retaining other forcings from the 2019–2020 period. This involved including the two post-monsoon cyclones in 2018. However, neither of them had penetrated north of 15°N (Fig. 1), thus having no discernible influence on the formation of ASHSW. The salinity distribution from the experiment resembled that from the 2018–2019 control experiment (Fig. 6), further confirming that salinity freshening in 2019–2020 was caused by cyclone induced precipitation.

**PPE$_{2019-23}$.** To isolate the impact of cyclone-induced precipitation on salinity freshening during the 2019–2023 period, we conducted a third perturbation experiment. In this experiment, referred to as PPE$_{2019-23}$, the model is initialized in January 2019 from the control run and integrated through December 2023. In the simulation, the 2018 precipitation forcing has persisted over the 2019–2023 period, while maintaining all other forcings intact. In retrospect, 2018 was characterized as a typical year, with cyclones exerting minimal influence on the formation of ASHSW. In this context it can be comparable to the 2014–2018 periods of typical ASHSW formation. Thus, by contrasting PPE$_{2019-23}$ with the control simulation, we can isolate the precipitation events that were responsible for the most significant observed salinity freshening during the period 2019–2023.

While the perturbation experiment PPE$_{2019-23}$ successfully isolates the contributions of precipitation anomalies relative to the typical year 2018, thereby mimicking the salinity anomalies between the periods 2014–2018 and 2019–2023, the separate influences of tropical cyclones and extreme precipitation events (as shown by the blue and green lines in Fig. 2c and the salt-budget analyses in Fig. 8c) on ASHSW freshening remain to be fully distinguished. Precipitation from tropical cyclones is typically short-lived, lasting a few days to weeks, whereas extreme precipitation events can persist for longer durations, often spanning months. Separating the relative contributions of these two rainfall sources is important, as their impacts on salinity freshening, differ in magnitude and duration.

**NOTC.** In the first experiment, referred to as NOTC, precipitation from all tropical cyclones during 2019–2023 is suppressed. While the ideal approach would be to suppress precipitation based on the storm radius, as done for the observations (Fig. 2c), discrepancy between NAVGEM precipitation and the observational data necessitated a simplified method. Precipitation is set to zero within the region 48°–78°E, 5°–25°N for the duration of each cyclone, with a ±2-day window to account storm remnants during the formation and post-landfall phases[16]. All other forcings remain consistent with the respective years. The model is initialized on January 1, 2019, from the CNTL run and integrated forward through 2023. As most pre-monsoon cyclones occur between May and early June and post-monsoon cyclones between September and October, the suppression of summer monsoon rainfall during these periods is minimal.

**NOPE.** To isolate the impact of extreme precipitation events in 2020 and 2022 on salinity freshening, a second experiment, NOPE, is conducted. For both events, precipitation is suppressed (set to zero) during July–August in the region east of 60°E and north of 15°N. This region is chosen due to its importance for ASHSW formation, where freshwater input influences stratification[7], while preserving summer monsoon precipitation along the west coast of India. As with the NOTC experiment, the model is initialized on January 1, 2019, from the CNTL run and integrated forward through 2023.

**NOPETC.** Finally, to evaluate the combined impact of tropical cyclones and extreme precipitation events, a third experiment, NOPETC, is conducted. In this experiment, precipitation is suppressed for both tropical cyclones and extreme events, following the same methods used in NOTC and NOPE. The model is again initialized on January 1, 2019, from the CNTL run and integrated forward 2023.

## Salinity budget in the Northern Arabian Sea

We take a more detailed look at the physical processes that control each term in the salinity budget. The salinity budget equation[25], can be written as:

$$\frac{\partial S}{\partial t} = S(E - P - R) - \boldsymbol{u} \cdot \boldsymbol{\nabla S} - \boldsymbol{w}\frac{\partial S}{\partial z} + F_h + F_v \qquad (4)$$

where $E$ is evaporation, $P$ is precipitation, $R$ is river runoff, $S$ is surface salinity, $\boldsymbol{u}$ is horizontal velocity components $(u, v)$ and $\boldsymbol{w}$ is vertical velocity, $\boldsymbol{\nabla S}$ is the horizontal derivative of salinity $(x, y)$. In more simple terms, we can express above equation as:

$$\Delta S_{totl} = \Delta S_{sflx} - \Delta S_{hadv} - \Delta S_{vadv} + \Delta S_{hmix} + \Delta S_{vmix} \qquad (5)$$

Where $\Delta S_{totl}$ is the salinity tendency, with terms on the right side representing various contributors. The term $\Delta S_{sflx}$ represents the salinity change due to surface freshwater fluxes (evaporation, precipitation, and runoff). The salinity tendency due to horizontal advection ($\Delta S_{hadv}$), vertical advection ($\Delta S_{vadv}$) accounts for the effect of advection. The term $\Delta S_{hmix}$ represents salinity change due to horizontal mixing, which accounts for unresolved small-scale turbulence and mixing processes. $\Delta S_{vmix}$ represents salinity changes due to vertical mixing processes, including wind- and buoyancy-driven mixing, entrainment, and turbulent salinity flux at the base of the mixed layer. Since the horizontal mixing term is negligibly small compared to other terms in the salinity tendency equation, it is excluded from the analysis.

These terms are computed at each time-step and accumulated over a 24-h period during the model simulations. The salinity budget is evaluated within the ASHSW formation region spanning 60°–70°E, 15°–25°N. Additionally, the terms are vertically averaged within the upper 100 m to capture the processes governing the formation of ASHSW (Fig. 8). Note that the salinity budget terms presented here focus on the processes controlling salinity variability, rather than closing the budget terms, which would require accounting for fluxes in and out of the control volume.

## Data availability

The Met Office Hadley Centre series of datasets (EN4) of global quality-controlled ocean temperature and salinity profiles are openly available at https://www.metoffice.gov.uk/hadobs/en4, also cited in Good et al.[45]. Argo float observations are publicly available at argo.ucsd.edu. Precipitation data (IMERG) can be accessed from gpm.nasa.gov and tropical cyclone track data (IBTrAC) from ncei.noaa.gov. Due to the large volume of model outputs, the data are available upon request to the corresponding author. Data to reproduce the figures in the article are provided at https://doi.org/10.6084/m9.figshare.28291862.

## Code availability

The code used to produce the figures in this study is available upon request from the corresponding author. All figures are generated using Pyferret, an open-source software freely available at https://ferret.pmel.noaa.gov/Ferret/.

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

## Acknowledgements

This work was supported by the U.S. Office of Naval Research through the NRL project "winter convection and water-mass formation modulated by mesoscale eddies (PE 61153N)". The computational resources were provided by the Department of Defense (DoD) High Performance Computing Modernization Program and the simulations were performed on the supercomputers at the Navy DoD Supercomputing Resources Center, Stennis Space Center, Mississippi. This is NRL contribution NRL/JA-7320-24-6511. It has been approved for public release and distribution is unlimited.

## Competing interests

The author declares no competing interests.
