## [Transparent Peer Review file · Nature Communications]

Sustained Freshening of Arabian Sea High Salinity Water Induced by Extreme Precipitation Events

Corresponding Author: Dr Prasad Thoppil

Version 0:

Reviewer comments:

Reviewer #1

(Remarks to the Author)

- What are the noteworthy results?

I had great pleasure in reading the manuscript entitled "Sustained Freshening of Arabian Sea High Salinity Water Induced by Extreme Tropical Cyclone Precipitation". The manuscript investigates the impact of freshening happened in northern Arabian Sea due to anomalous number of Tropical Cyclones (TCs) happened during 2019 -2023 by contrasting them with the freshening due to Tcs during 2014-2018 period. The author used, in situ Argo float measurements, satellite datasets and model simulations to examine the contrasts in freshening and resulting changes in Arabian Sea High salinity water mass formation during the two periods selected for the study.

Important results:

- For the first time there is an attempt to show the impact of TC induces precipitation on the upper ocean water-mass formation and salt-budget of the northern Arabian Sea
- The authors use both observations and model simulations and demonstrate that both agree in demonstrating the impact of TC induced precipitation.
- This study encourages to investigate future trends TC trends in northern Arabian sea and the possible impact on water-mass formation which has widespread impact on ecological balance of the region.

- Will the work be of significance to the field and related fields? How does it compare to the established literature? If the work is not original, please provide relevant references.

This work is certainly novel and can give potential insights about the impact of TC associated rainfall on water-mass formation which has widespread influence on ecology, fisheries and productivity in the region.

- Does the work support the conclusions and claims, or is additional evidence needed?

The present analysis shown by the author supports the conclusions and claims to a good extend. However, It will be interesting if the author can clarify below questions which can improve the confidence on this work further.

Arabian sea is a basin under the influence of Indian monsoon and there is strong seasonality for rainfall in this region. So naturally there is existence of freshening and evaporation cycle in this region. It will be interesting to understand if the differences in amplitude of freshening due to Tcs during 2019-2023 is statistically significant with the historical amplitude of freshening due to the impact of monsoonal rainfall.

- Are there any flaws in the data analysis, interpretation and conclusions? Do these prohibit publication or require revision?

The data analysis approach is sound in general, however, author may clarify below aspects.

The author show differences in mean potential energy anomaly during 2024-2018 and 2019-2023. Is this difference in mean statistically significant? Can this be verified with a statistical significance test?

- Is the methodology sound? Does the work meet the expected standards in your field?

Authors use best possible datasets and model simulations to prove their arguments.

And it meets the standards. The minor issues are pointed out under other relevant headings.

- Is there enough detail provided in the methods for the work to be reproduced?

Author use two quantities in the manuscript potential energy anomaly and ASHSW volume. Are there references for these quantities? Please specify.

The datasets used in the study is available from public domain, however author may be asked to make the model simulations available if the results need to be reproduced using model simulations as shown in the manuscript

Minor comments:

1) On page 4 towards the end (it may differ as I checked word document) Although no cyclones occurred in 2022, the extremely high rainfall during July – August, which caused severe flooding and displaced over 30 million people in Pakistan, helped maintain one of the largest salinity anomalies during 2019 – 23 ever recorded (Figure 3).

Here a reference may be added, if author wanted to state it is largest salinity anomalies “ever” recorded. Else they may modify saying largest salinity anomalies during the study period.

2) On page 7 line two, author says “ We estimated Potential Energy Anomaly” which may be re-written as I estimated as it is a single author paper or the contributor may be acknowledged if there is one.

Reviewer #2

(Remarks to the Author)

My review is detailed in the attached pdf documents below

[Editorial Note: This attachment is displayed in the final 12 pages of this file]

Reviewer #3

(Remarks to the Author)

The manuscript “Sustained Freshening of Arabian Sea High Salinity Water Induced by Extreme Tropical Cyclone Precipitation” was reviewed. The paper seems to address an interesting and important issue in these less studied and important ocean basin. It also appears to entail a substantial effort. Hence it can deserve being published. However, some comments listed below may improve it:

1- The text is well written, but repetitions of some text parts may be reduced (changed) as they are noticeable and can be a bit tedious to the readers? As: “cyclone-induced”, “formation of ASHSW” or “ASHSW formation”, “salinity freshening”. If it can be helped?

2- Moving Figs to be more harmonious with text may be useful!?

3- The present work only presents salinity and precipitation records and distributions, and no records of surface heat (especially temperature records from observations or model) and momentum (currents?) fluxes are mentioned (even from the work of the others?) as they are important in understanding the issue being addressed here?

4- No cyclone precipitation in 2022 but systemic (synoptic, as large size cyclone or depression) precipitation is substantial!, but the cyclone-induced precipitation is often emphasized? Systemic precipitation have also contributed in other years and cyclone-induced ones are also short lived! Shifting pattern of precipitation northwards on the whole may be more important? Therefore, the paper title may be changed?

ITCZ in summer, appear to have shifted more (it usually change longitudinal position much more in Indian ocean than in other equatorial regions during the year, hence monsoonal systems) northwards in recent years in the Indian Ocean region that may be due to recent fast climate change, this could have increased the overall precipitation in the Arabian sea?[Mamalakis, A., Randerson, J.T., Yu, J.Y. et al. Zonally contrasting shifts of the tropical rain belt in response to climate change. Nat. Clim. Chang. 11, 143–151 (2021).]

5- In Fig 3 they may present temperature records along with shown salinity anomaly, and temperature profile in Fig.2, to

show better the density stratification which is claimed to have been “substantially weakened”? What is the cooling effects of cloudiness during precipitation? Temperature records may show this. Although in Fig. 4 potential density shows a decrease, along with the shallowing mixed layer depth. (E.g. paper by Shee et al. Recent changes in the upper oceanic water masses over the Indian Ocean using Argo data, *Sci. Rep.* 2023, 13, 20252, also show some temperature records that may be worth looking at?). So presentation of temperature profiles and series changes may be recommended.

6- As the northern Indian Ocean is experiencing the intrusions of high salinity and temperature at depth of about 100 to 500 m from the Persian Gulf and Red Sea outflows (depending on season) while being refreshed by the surface water (especially the Persian Gulf, which is also refreshed by the surface water of Arabian Sea), have these semi-enclosed seas any influence on the ASHSW? It may be worth commenting on this, in discussion?

7- Arabian Sea oxygen- depleted waters and hence, fish kill has increased recently, can ASHSW changes influence this; this may be interesting to just point out? Oxygenation of the Oman Sea and northern Indian Ocean waters by these outflows (4) has been shown to be important, although they are at deeper depths (e.g. Font, E., Swart, S., Bruss, G., Sheehan, P. M. F., Heywood, K. J., & Queste, B. Y. (2024). “Ventilation of the Arabian Sea oxygen minimum zone by Persian Gulf water”. *Journal of Geophysical Research – Oceans*, 129(5), [e2023JC020668]).

8- Fig. 1 show cyclone track shifted northwards, is this due to recent climate change? Not clearly reasoned? Precipitation seems to have increased more over the eastern coast of the Arabian Sea (more summer ITCZ shift, 4, ?).

9- Should $\Sigma \Delta P$ be + and $\Sigma \Delta S$ be - in Fig. 3(c)? Can these be decadal Oscillation?

10- In salinity budget equation, F_v and F_h are probably vertical and horizontal fluxes due other forcing (may be turbulence or eddy fluxes)? They better to be introduced clearly.

Best of luck for the author.

Version 1:

Reviewer comments:

Reviewer #1

(Remarks to the Author)

The author has addressed the concerns raised by me and I suggest the manuscript may be published.

Reviewer #3

(Remarks to the Author)

The author has made a major effort in improving the paper following our comments on its early version. Hence I recommend the paper for publication.

REVIEWER COMMENTS

Reviewer #1 (Remarks to the Author):

- What are the noteworthy results?

I had great pleasure in reading the manuscript entitled “Sustained Freshening of Arabian Sea High Salinity Water Induced by Extreme Tropical Cyclone Precipitation”. The manuscript investigates the impact of freshening happened in northern Arabian Sea due to anomalous number of Tropical Cyclones (TCs) happened during 2019 -2023 by contrasting them with the freshening due to Tcs during 2014-2018 period. The author used, in situ Argo float measurements, satellite datasets and model simulations to examine the contrasts in freshening and resulting changes in Arabian Sea High salinity water mass formation during the two periods selected for the study.

Important results:

- For the first time there is an attempt to show the impact of TC induces precipitation on the upper ocean water-mass formation and salt-budget of the northern Arabian Sea
- The authors use both observations and model simulations and demonstrate that both agree in demonstrating the impact of TC induced precipitation.
- This study encourages to investigate future trends TC trends in northern Arabian sea and the possible impact on water-mass formation which has widespread impact on ecological balance of the region.

We sincerely thank the reviewer for their thoughtful comments and constructive suggestions, which have greatly enhanced the quality of the manuscript and strengthened the scientific arguments presented. We have carefully addressed the majority of the reviewer’s comments and made the necessary modifications to the manuscript. Below, we provide a detailed point-by-point response, with the reviewer’s comments in black and our responses highlighted in blue. We hope that these revisions adequately address the reviewer’s concerns.

- Will the work be of significance to the field and related fields? How does it compare to the established literature? If the work is not original, please provide relevant references.

This work is certainly novel and can give potential insights about the impact of TC associated rainfall on water-mass formation which has widespread influence on ecology, fisheries and productivity in the region.

- Does the work support the conclusions and claims, or is additional evidence needed?

The present analysis shown by the author supports the conclusions and claims to a good extend. However, It will be interesting if the author can clarify below questions which can improve the confidence on this work further.

Arabian sea is a basin under the influence of Indian monsoon and there is strong seasonality for rainfall in this region. So naturally there is existence of freshening and evaporation cycle in this region. It will be interesting to understand if the differences in amplitude of freshening due to Tcs during 2019-2023 is statistically significant with the historical amplitude of freshening due to the impact of monsoonal rainfall.

The reviewer raises an excellent point regarding the potential influence of seasonal and interannual variability on precipitation, particularly from the Indian Monsoon and other climate drivers such as the Indian Ocean Dipole (IOD) and Monsoon Intra-Seasonal Oscillations (MISO). Indeed, the precipitation anomaly shown in Figure 2b (after removing the monthly climatology) reveals substantial anomalies even during non-cyclone periods, indicating interannual variability in monsoonal rainfall.

To address the reviewer's concerns, we have expanded the analysis in the revised manuscript as follows:

1. **Separation of Precipitation Sources:** We distinguished precipitation contributions from tropical cyclones and other extreme precipitation events using IMERG observations, as shown in the updated Figure 2c.
2. **Additional Model Experiments:** We performed three additional model experiments designed to quantify the separate and combined impacts of tropical cyclones and extreme precipitation events (notably in 2020 and 2022) on salinity freshening during the 2019–2023 period (new Figure 9).

These results indicate that the majority of the observed salinity freshening during 2019–2023 can be attributed to the combined effects of tropical cyclones and the extreme precipitation events in 2020 and 2022. The contribution from broader monsoonal rainfall variability or other interannual climate modes appears minimal in comparison, further emphasizing the dominant role of these extreme events in driving the observed salinity freshening.

While the above approach differs slightly from the reviewer's suggestion, it effectively addresses the underlying concern. To further clarify, the differences in precipitation due to tropical cyclones during 2019–2023 can be derived from IMERG observations (Figure R1a). By subtracting the long-term mean (2001–2023) from the 2019–2023 mean, we obtain the precipitation anomaly for this period (Figure R1b). By excluding the contributions from tropical cyclones (Figure R1c) and the combined cyclones and extreme precipitation events of 2020 and 2022 (Figure R1d) from the 2019–2023 mean, further illustrate the influence of monsoon rainfall, which is small. The methodology used to separate precipitation from cyclones and extreme events is included in Methods. Since this analysis has already been incorporated into the revised manuscript (Figure 2c), we have opted to include this additional breakdown in Supplementary Figures 1 and 2.

Figure R1. Precipitation differences (mm d^{-1}) highlighting the contributions of tropical cyclones and extreme precipitation events during 2019–2023: (a) Precipitation anomalies directly associated with tropical cyclones during 2019–2023. (b) Difference between the 2019–2023 mean precipitation and the long-term average (LTA, 2001–2023). (c) Difference between the 2019–2023 mean precipitation excluding tropical cyclones and the LTA, isolating non-cyclone contributions. (d) Difference between the 2019–2023 mean precipitation excluding both tropical cyclones and extreme precipitation events of 2020 and 2022 and the LTA, highlighting background precipitation patterns (see Methods for separation of precipitation contributions from tropical cyclones and extreme events).

- Are there any flaws in the data analysis, interpretation and conclusions? Do these prohibit publication or require revision?

The data analysis approach is sound in general, however, author may clarify below aspects. The author show differences in mean potential energy anomaly during 2024–2018 and 2019–2023. Is this difference in mean statistically significant? Can this be verified with a statistical significance test?

Thank you for the insightful suggestion. In response, we have conducted statistical significance testing on the differences in means and have included error bars in Figure 4 to reflect the associated uncertainty. The results indicate that the differences in means are statistically significant at the 95% confidence interval ($p = 0.05$).

- Is the methodology sound? Does the work meet the expected standards in your field?

Authors use best possible datasets and model simulations to prove their arguments. And it meets the standards. The minor issues are pointed out under other relevant headings.

- Is there enough detail provided in the methods for the work to be reproduced?

Author use two quantities in the manuscript potential energy anomaly and ASHSW volume. Are there references for these quantities? Please specify.

Thank you for the suggestion. The reference for potential energy anomaly has cited in the main text (23), as well as in Methods section, alongside the additional reference (43). Furthermore, the use of ASHSW volume in the Arabian Sea, as previously used by (8), has also been incorporated into the Methods section.

The datasets used in the study is available from public domain, however author may be asked to make the model simulations available if the results need to be reproduced using model simulations as shown in the manuscript

Yes, the model simulation experiments will be made available to interested researchers.

Minor comments:

1) On page 4 towards the end (it may differ as I checked word document) Although no cyclones occurred in 2022, the extremely high rainfall during July – August, which caused severe flooding and displaced over 30 million people in Pakistan, helped maintain one of the largest salinity anomalies during 2019 – 23 ever recorded (Figure 3).

Here a reference may be added, if author wanted to state it is largest salinity anomalies “ever” recorded. Else they may modify saying largest salinity anomalies during the study period.

Thank you for pointing this out. We have revised the sentence to: “Although no cyclones occurred in 2022, the extremely high rainfall during July–August, causing severe flooding and displacing over 30 million people in Pakistan¹⁷, contributed to sustaining salinity freshening through 2023”.

2) On page 7 line two, author says “ We estimated Potential Energy Anomaly” which may be re-

written as I estimated as it is a single author paper or the contributor may be acknowledged if there is one.

As suggested by the reviewer, the sentence is re-written. Thank you.

Reviewer #2

Summary:

In this MS, the author first analyzes the impact of Tropical Cyclones (TCs)-induced precipitation on upper ocean salinity in the North Arabian Sea (AS) over 2014-2023 using satellite rain and in situ Argo profile data. The author reports an increased number of TC in 2019-2023, concomitant with an increase in total precipitation and an overall decrease of upper ocean salinity in the East AS with respect to the 2014-2018 period, with a maximum freshening observed in the North East AS region. He then analyzes the temporal evolution of the vertical salinity structure using in situ profile data, revealing sustained freshening in the Northern Arabian Sea where the Arabian Sea High Salinity Waters (ASHSW) form. The upper ocean freshening reduces the potential surface density, augmenting the vertical stratification and in fine, disrupt the convective sinking process of ASHSW. Finally, using model perturbation experiment and model-based salinity budget estimates, the author attempts to isolate the impact of TC-induced rain on salinity freshening and conclude that the recent SSS freshening is caused by the increased number of TCs from 2019 onward.

We sincerely thank the reviewer for their thoughtful comments and constructive suggestions, which have greatly enhanced the quality of the manuscript and strengthened the scientific arguments presented. We have carefully addressed the majority of the reviewer's comments and made the necessary modifications to the manuscript. Below, we provide a detailed point-by-point response (reviewer comments in black and our responses highlighted in blue). We hope that these revisions adequately address the reviewer's concerns.

While the revised manuscript with track changes is included, the major revisions addressing the reviewers' concerns are outline below under subsections:

- Prolonged Salinity Freshening: Observations: Detailed revisions can be found in paragraphs 2–5, where we have separated precipitation contributions from cyclones and other extreme events.
- Isolating Precipitation Sources Driving ASHSW Freshening: Results from model experiments incorporating those precipitation sources contributing to the freshening of ASHSW are provided in paragraphs 1–4.

Major Comments and recommendations

The topic is of strong scientific interest, in particular for the tropical cyclone forecast, salinity remote sensing, and, ocean modeler communities. While the SST, CHLA, and SSH response to TC passage have been extensively studied, TC-induced SSS changes and their large-scale impacts on other climate processes are much less known: this work definitively participate to these community efforts. In general, the MS reads well, is interesting and the results are in general plausible. However, I have several major concerns.

First, in the part of the MS devoted to observations analyses, the author had not at all isolated the TC-induced precipitation and associated SSS changes from other sources of

atmospheric and oceanic variability, which weaken strongly the strength of his statements, which are often more ‘suggestions’ than a clear demonstration of causality between the three reported interannual anomalies seen in the observation: 1) ‘increased number of TC’, 2) ‘increased total Precipitation’, and 3) ‘decreased overall upper ocean Salinity’. For me, the author has not demonstrated convincing evidences from his analyses that the reported large-scale upper ocean salinity and precipitation anomalies (shown in Figure 1 & Figure 2 in the MS), as well as their consequences on the stability of the water column (Figure 4) is dominantly due to the passage of some rare but intense tropical cyclones rainfall and associated SSS freshening. I will show below in the detailed comments section of my review several examples in 2019 where there was little precipitation accumulated during the passage of some strong TCs in this region as well as small associated SSS changes in their wake, as reported from satellite observations. The interannual SSS & P changes observed by the author could therefore be due to other processes (see details in Behara et al, 2019) such as the increased precipitation during the recent Monsoon, or e.g., due to increased runoff from the river waters out of the Gulf of Khambhat estuaries, or from elsewhere in the coasts (e.g Pakistan floods in 2022 as mentioned by the author at the end of his MS), or even more likely by large scale climate variability impact on rain and SSS (e.g., Indian Ocean Dipole, Moonsoon IntraSeasonal Oscillation (MISO),...) and not necessarily because of changes in Tropical Cyclone activity per se in between 2019-2023 and 2014-2018. For example, Li et al (2022) found that the variance of MISO has an increasing trend from 1982 to 2017 over the northeastern Arabian Sea (AS), accompanied by increasing intraseasonal rainfall. How does this impact the precipitation excess shown in Figure 1c? The enhancement in rainfall in this case is mainly nourished by the increasing moisture supply, with a major contribution from the upward moisture advection by the intraseasonal vertical wind velocity. Since the author did not isolate the precipitation signal from TC-only with respect to total rain, and also did not isolate the salinity response to TC passage, most of his statements remain poorly evidenced, while this can be now easily estimated from the observation dataset already gathered by the author:

- 1) From IMERG and Ibracs datasets, the author could estimate the rain accumulated during the passage of each storm over 2014-2023 in a spatial domain (e.g. within the wind radius of 34knot winds or within a radius of 500 km) around the TC center tracks.
- 2) Then further estimate the time series of the cumulated contribution of tropical cyclone only induced rainfall to the total precipitation over the 3 AS subregions selected by the author. Then he could compare the relative contribution of TCs separately for both period 2014-2018 and 2019-2023.

The respective contribution of TC-induced precipitation to total rain and their interannual anomalies could then be much better estimated and would strongly re-enforce the author’s statements. Such approach has been used frequently in the past literature based on satellite rain products and could be applied following for example the approaches proposed in Lonfat et al (2004), Chen and Fu (2015), Jiang and Zipser (2010), Jiang et al (2011). For example, a recent study by Camberlin et al. (2024) evaluated the contribution of the Western Arabian Sea Tropical Cyclones to Rainfall in the Horn of Africa and Southern Arabian Peninsula by estimating rainfall on non-tropical cyclone (TC) days and on TC days based on IMERG data (2000–2020).

Thank you very much for your constructive comments and suggestions, which have greatly improved the quality of the manuscript. We particularly appreciate the extensive and detailed overview of related studies and the insightful recommendations that have strengthened the scientific arguments presented in the paper. We are especially grateful for the time and effort you dedicated to providing relevant data analyses and examples that further support this study. While the review was thorough and comprehensive, which we highly value, our responses focus primarily on addressing the key concerns raised and clarifying how they have been incorporated into the revised manuscript.

To address the reviewers' primary concern regarding the separation of cyclone-induced precipitation from the total precipitation, we have followed the reviewer's recommendations and the methodology described by Camberlin et al. (2024). Specifically, we separated cyclone-induced precipitation from the total precipitation, and the cumulative precipitation anomalies excluding cyclone-induced rainfall during the 2019–2023 period are now presented in Figure 2c (previous Figure 3c).

Furthermore, we extended this analysis to distinguish extreme precipitation events, which significantly contributed to the salinity freshening observed in 2022. Using the 95th percentile threshold, we identified and isolated extreme precipitation events. Our analysis revealed that both 2022 and 2020 experienced extreme precipitation events, although the 2020 event coincided with two tropical cyclones. The combined cumulative precipitation anomalies, excluding both cyclone-induced rainfall and extreme precipitation events, are also depicted in Figure 2c.

We also acknowledge that the increasing trend in extreme precipitation events over the Arabian Sea may be linked to large-scale climate variability, including the IOD and MISO. Notably, MISO-related rainfall has substantially intensified over the northeastern Arabian Sea in the past three decades, driven by enhanced Arabian Sea warming (Li et al. 2022). This aspect has been incorporated in the "Discussion section".

We have repositioned Figure 3 and the associated discussion earlier in the manuscript, following Figure 1. This restructuring enables for a clearer definition of precipitation anomaly sources and provides a more logical framework for explaining ASHSW freshening in subsequent sections, focusing on the role of precipitation events. The previous Figure 3 has now been numbered as Figure 2.

Although dividing the precipitation analysis into three subregions of the Arabian Sea seems logical, it may not effectively explain ASHSW freshening, as ASHSW changes are not purely local but also influenced by its subsequent spreading. Therefore, we focus on the eastern Arabian Sea as a whole for the observational analysis. However, in the modeling experiments, we separately analyze the northern Arabian Sea, where ASHSW forms, and the eastern Arabian Sea, where the most significant freshening has occurred.

Lastly, the reviewer correctly highlights that local changes in sea surface salinity (SSS) during cyclones can be assessed by comparing pre- and post-storm SSS, as shown in the reviewer's Figure 2. However, while this approach effectively captures the immediate salinity response to a cyclone, it does not reflect the delayed and cumulative effects on ASHSW formation and spreading. ASHSW formation is primarily driven by excess evaporation over precipitation, and the recent increase in freshwater input from tropical cyclones and extreme precipitation events

has altered this balance, leading to stronger stratification in the northern Arabian Sea. Since this region typically experiences net evaporative cooling throughout the year, the northward shift of tropical cyclones and increased extreme precipitation have contributed additional freshwater, further preconditioning the area for sustained ASHSW freshening. Our study focuses on this evolving freshwater imbalance and its influence on ASHSW formation, which cannot be fully captured by short-term SSS comparisons alone.

Similarly, to best isolate the TC-induced surface freshening from other sources of SSS variability (e.g., Moonsoon rain, horizontal advection, river runoff impacts), **the SSS change induced by each TC can now be evaluated systematically with confidence using satellite SSS data** back to 2010 following the approaches presented in numerous papers in the literature (see for examples Chaudhuri et al., 2019; Grodsky et al., 2012; Neethu, 2018; Reul et al., 2014; Yue et al., 2018, Reul et al. 2021 and Sun et al., 2021). Merged SMOS, Aquarius and SMAP high quality Level 4 satellite SSS products now exist to perform such analyses (e.g., ESA Climate Change Initiative SSS products, Boutin et al. 2021 or the Multi-mission L4 Optimally Interpoated Sea Surface Salinity, Melnichenko, 2021). **The author could then evaluate the cumulated TC-induced SSS changes** in each season over 2014-2023 and see how much it contributed to the overall SSS decrease observed in that region since 2019.

Thank you again for this excellent point and the detailed explanation. While our approach differs from the method proposed by the reviewer, we have addressed the concern using model experiments. Below, we explain how we addressed the issue and clarify why the reviewer's suggested method may not be fully suitable for this study.

To isolate the impact of precipitation from tropical cyclones and extreme precipitation events (2020 and 2022) on salinity freshening during the 2019–2023 period, we conducted three additional model experiments by selectively excluding: (1) cyclone-induced precipitation, (2) extreme precipitation events in 2020 and 2022, and (3) the combined effects of both. We then compared these experiments against a control simulation that included all precipitation forcings. These experiments confirmed that the combined influence of cyclone-induced precipitation and the two extreme precipitation events accounted for a significant portion of the salinity freshening observed during the 2019–2023 period.

We agree that satellite-derived sea surface salinity (SSS) can be useful for estimating localized, short-term salinity changes induced by tropical cyclones, as demonstrated in several studies, including the example provided by the reviewer. However, satellite SSS is less suitable for the objectives of this study due to the following reasons:

- Our focus is on the long-term impact of precipitation on the formation and seasonal spreading of the Arabian Sea High Salinity Water (ASHSW), which occupies the upper 100 m of the water column.
- While SSS changes during a cyclone can be inferred by comparing pre- and post-storm SSS (as illustrated in the reviewer's Figure 2), it does not capture the delayed and prolonged effects on ASHSW formation and spreading. For instance, freshwater input from the extreme

precipitation event in June 2019 could weaken the formation of ASHSW the following winter (November 2019–February 2020), with its influence observed later in the summer monsoon season (August–October 2020) as the ASHSW spreads southward.

- The vertical salinity sections provided in the manuscript (at three locations) reveal the vertical structure and extent of ASHSW, which cannot be inferred from satellite SSS alone. For this reason, the satellite SSS provides limited additional insight compared to the Argo profile observations already presented in Figure 3.

On the other hand, the model experiments presented in the manuscript were specifically designed to quantify the impact of precipitation on ASHSW formation and subsequent spreading. These controlled simulations allow us to isolate processes that cannot be fully captured using observations alone.

Second, in the second part of the MS devoted to model analyses, here again the author has not directly isolated the impact of interannual variability of the TC-induced precipitation and associated SSS changes from other sources of interannual atmospheric and oceanic variability. While I agree with the author that by inverting atmospheric forcing of years 2018 and 2019 the model can reproduce the SSS freshening observation: it does only show that the interannual variability in P predominantly modulate the SSS signal: the author however did not show directly that the TC-induced rain only is responsible for that interannual signal: some other sources of precipitation variability over 2018-2019 could be responsible for it. Once the TC-induced rain will be isolated from the total precipitation time series as I suggested previously, the author could then easily re-run his model perturbation experiments by only removing TC days in the atmospheric forcing time series: this would directly show what the author want to prove rather than by using lengthy and sometimes confusing indirect approaches.

Thank you again for the insightful suggestions. The additional model experiments, with and without cyclone-induced precipitation, presented in Figures 9d and 9e, effectively illustrate the impact of tropical cyclones on salinity freshening. Specifically, in the northern Arabian Sea, the control experiment shows a fresher salinity by approximately 0.2 psu during November – December 2019 (Figure 9e) compared to the experiment where cyclone-induced precipitation was excluded for the same period.

The primary objective of this case study, focusing on the 2018–2019 period, is to demonstrate two key processes:

1. How precipitation from post-monsoon tropical cyclones during the September–December period influences the formation of ASHSW north of $\sim 15^{\circ}\text{N}$ (the ASHSW formation region).
2. The subsequent southward spreading of ASHSW along the eastern Arabian Sea during the following late summer (September–October 2020).

The current model setup allows us to compare the resulting salinity patterns to those from the reference control run. While the observed salinity differences in these experiments are attributed

to variations in precipitation during the 2018–2019 period, as the reviewer correctly pointed out, we acknowledge that precipitation during September–December in the northern Arabian Sea is generally minimal in a typical period. Thus, the precipitation pattern observed during September–December 2018 can be considered representative of typical conditions. In contrast, the precipitation from five tropical cyclones during September–October 2019 significantly contributed to the weakening of ASHSW formation and its subsequent spreading along the eastern Arabian Sea.

In addition, many time in the MS, the author used very strong statements (particularly on the impacts of TC-precipitation on SSS anomalies) without full proof of them: please rephrase those sentences to only state what is shown by your results.

In the revised manuscript, we have made efforts to avoid strong statements of certainty. While the initial part of the discussion primarily focuses on observations, the interpretation of these results remains largely qualitative. As the processes governing ASHSW anomalies become clearer in the latter sections, particularly with the additional analysis of model results quantifying these processes, the language has been adjusted to reflect this progression more accurately.

Therefore, despite most of the results shown by the author are plausible, I recommend that the MS undergo major revision before being considered for publication so that the author actually and properly separate the TC impacts from other sources of variability in both the Precipitation and SSS signals and in the observation and model analyses, separately: this shall in fine directly demonstrate the authors indirect statements.

Details are given below.

Detailed Comments:

L38 “evaporation-precipitation”- What about river runoff and their impacts on the northeast Arabian Sea (AS) upper salinity, in particular from the Indus and the Narmada, Sabamati, Mahi and Tapi rivers in the Gulf of Khambhat estuaries ? As shown in Rao et al (2009) and reproduced here below (panel d), rivers have non negligible impacts on the SSS in the North eastern AS, particularly in winter months.

[REDACTED]

Figure 1: Monthly climatological salinity for (a) July, (b) October, (c) December and (d) ARGO observations averaged for 17–20N during 2004. From Rao et al (2009)

Thank you for highlighting this relevant study. We agree with its findings. However, the river runoff in the northeastern Arabian Sea has a negligible impact on the formation and seasonal spreading of ASHSW, as the low-salinity water associated with river discharge remains largely confined to shallow regions along the west coast of India. While it is plausible that an increased frequency of cyclones and extreme precipitation events could lead to enhanced river runoff and contribute to surface salinity freshening in these coastal regions, the model experiments presented in our study explicitly exclude such contributions, further supporting the critical role of precipitation forcing. Although the influence of river runoff is an important topic worth further investigation, it falls beyond the scope of the present study. A brief description of river runoff treatment in the model has been added to the Methods section for clarity.

L107-108: this sentence is a very strong statement that definitively need more support than what is shown and discussed around Fig1 at this stage in the MS. Please re-phrase to be more accurate and keep to the point of what your data really show. It is not because Fig1 panels reveal a concomitant increase in 1) the number of TCs in the North-eastern part of the AS, 2) as well as in overall Precipitation in the eastern AS and 3) a decrease in SSS in the NE AS that this is clearly an evident proof that “the cyclone-induced precipitation have impact on the convective formation of the Arabian Sea High Salinity Waters “. This is a very strong shortcoming. As I will show further, TC in the Arabian sea are not systematically associated with heavy rain and SSS drops are not systematically observed in their trail. Deeper observation analyses are required to conclude.

Indeed, the increase of precipitation shown in Fig1c could be due to other atmospheric sources of interannual variability (e.g., Indian Ocean Dipole, Moonsoon) than merely the change in Tropical Cyclone activity in between 2019-2023 and 2014-2018. What about a potentially increase in monsoon season accumulated rain in the most recent years ?

Similarly, the SSS decrease in the AS, while I agree is most intense where most new TCs tracks where located in 2019-2023 is also a region of significant runoff from the Narmada, Sabamati, Mahi and Tapi rivers in the Gulf of Cambay estuaries. Could the reported interannual variability in SSS in this region be also related to increased river runoff in 2019-2023 ? and if yes, how much does it contribute to the observed SSS decrease with respect to the impact of local rain increases ?

To resolve these uncertainties, I suggest the author shall re-analyse the datasets as follows:

- 3) From IMERG and Ibracs datasets, estimate the rain accumulated during the passage of each storm over 2014-2023 in a spatial domain (e.g. within the wind radius of 34knot winds) around the TC center track.
- 4) Then further estimate the average cumulated spatial contribution of all tropical cyclone rainfall to total precipitation over the AS region, separately for both period 2014-2018 and 2019-2023.

The respective contribution of TC-induced precipitation to total rain and their interannual anomalies could then be much better estimated and would strongly re-enforce the author' statements. Such approach have been used frequently in the past literature based on satellite rain products and could be applied following for example the approaches proposed in Lonfat et al (2004), Chen and Fu (2015), Jiang and Zipser(2010), Jiang et al (2011). For example a recent study by Camberlin et al. (2024) evaluated the contribution of the Western Arabian Sea Tropical Cyclones to Rainfall in the Horn of Africa and Southern Arabian Peninsula by estimating rainfall on non-tropical cyclone (TC) days and TC days based on IMERG data (2000–2020). Such an approach could be easily applied by the author.

- 5) Similarly, to best isolate the TC-induced surface freshening from other sources of SSS variability (Moonsoon rain, advection, river runoff impacts), the SSS change induced by each TC can now be evaluated systematically with confidence using satellite SSS data back to 2010 following the approaches presented in numerous papers in the literature (see for examples Chaudhuri et al., 2019; Grodsky et al., 2012; Neethu, 2018; Reul et al., 2014; Yue et al., 2018, Reul et al. 2021 and Sun et al., 2021). Merged SMOS, Aquarius and SMAP high quality Level 4 satellite SSS products now exist to perform such analyses (e.g., ESA Climate Change Initiative SSS products, Boutin et al. 2021 or the Multi-mission L4 Optimally Interpoated Sea Surface Salinity, Melnichenko, 2021)

In fact, databases including TC-induced Precipitation & SSS changes for each individual TC globally over 2010-2020 were already developed in the frame of the ESA Marine Atmosphere eXtreme Satellite Synergy (MAXSS) project (see <https://www.maxss.org/>) and could be used by the author to better demonstrate his statements. This dataset is publicly available (doi:10.12770/cc0577e4-55d6-4aa9-a938-b4965be121ab, Reul et al., 2023), and provides, for each storm tracks from IBTrACS during the period 2010-2020, three types of storm-related

parameters: (1) the pre-storm upper ocean conditions (e.g., SST, SSS, upper ocean vertical density stratification strength, etc.), (2) the atmospheric forcing during the storm passage (maximum wind, accumulated rain and evaporation, wind power index, etc.) during the storm, as well as (3) the oceanic wakes left after the storm for an ensemble of key variables (SST, SSS, sea level height, ocean color parameters). I show here below few examples of estimated SSS anomalies (derived from the ESA CCI SSS products) left after TC passages in the AS as well as the corresponding accumulated precipitation (derived from merged satellite TRMM products) during their lifetime over the AS in 2019 and which is available in the MAXSS database. As evidenced from satellite data, TC passage in the AS in 2019 are not systematically associated with strong local SSS decreases and heavy precipitation: see e.g. examples of the extremely severe cyclonic storm MAHA in 2019 with small amount of accumulated precipitation associated with negligible SSS changes in the North Eastern AS, or the other case of the super cyclonic storm KYARR with locally very high accumulated rain at 20°N but no significant associated SSS changes observed from satellites. Significant SSS drops were nevertheless observed after the passage of the very severe cyclonic storm VAYU in a region with significant accumulated rain. Therefore, given the high variability of the SSS response to TC passages in this region, and although Sun et al., (2021) have shown (see their Figure 16 top panel) that on average TC do induce SSS drops in the AS, it remains unclear what the overall contributions of the TC-induced rain and associated freshening are on the 2014-2018 versus 2019-2023 interannual variability reported in Fig 1.c and Fig.1d.

Figure 2: Examples of SSS anomalies left after TC passages in the Arabian Sea in 2019 (left panels) and corresponding accumulated rain (right panels). Data and plots are available at <https://www.maxss.org/>.

Thank you for your time and effort in providing such extensive comments and a detailed explanation of methods along with several relevant references. The reviewer has raised a number of important points and provided valuable suggestions for strengthening the manuscript. Since many of these comments align closely with previous feedback, albeit in a more detailed manner, we would like to focus our responses on the primary concerns most relevant to the core scope of our study: demonstrating that precipitation from tropical cyclones and extreme precipitation events has been a significant driver of salinity freshening in the eastern Arabian Sea during the 2019–2023 period.

We greatly appreciate the reviewer’s independent analyses as part of this review and agree with the conclusions drawn from those efforts. However, as noted in previous responses, certain limitations exist regarding the types of data and approaches that can be applied within the scope of this study. For example, while the above mentioned analysis tool is useful for estimating local changes in sea surface salinity (SSS) or SST in the wake of TCs compared to pre-storm conditions, it does not directly capture the impact of precipitation on ASHSW formation and its subsequent spreading.

To address the reviewer's primary concern regarding the separation of cyclone-induced precipitation from total precipitation, we have now followed the reviewer's recommendations and the methodology described by Camberlin et al. (2024). Specifically, we separated cyclone-induced precipitation from total precipitation, and the cumulative precipitation anomalies excluding cyclone-related rainfall during the 2019–2023 period are now presented in Figure 2c.

Additionally, we extended this analysis to better differentiate extreme precipitation events, which significantly contributed to the observed salinity freshening, particularly in 2022. By applying a 95th percentile threshold, we identified and isolated extreme precipitation events. Our analysis revealed that both 2022 and 2020 experienced extreme precipitation events, although the 2020 events coincided with two tropical cyclones. The combined cumulative precipitation anomalies, excluding both cyclone-induced rainfall and extreme precipitation events, are now depicted in Figure 2c for clarity.

To further quantify the individual and combined impacts of precipitation from tropical cyclones and extreme precipitation events (2020 and 2022) on ASHSW freshening during the 2019–2023 period, we conducted three additional model experiments where we selectively excluded: (1) cyclone-induced precipitation, (2) extreme precipitation events in 2020 and 2022, and (3) the combined effects of both. These were compared against a control simulation that included all precipitation forcings. The results confirmed that the combined influence of cyclone-induced precipitation and extreme precipitation events accounted for a substantial portion of the salinity freshening observed during the study period.

Line 111-113: I agree that the addition of freshwater reduces surface density but it has not been demonstrated at this stage of the MS that it disrupts the sinking process and weakens the formation of the ASHSW. Please re-phrase to only state what is shown by your results.

As suggested by the reviewer, this statement has been moved up in the manuscript.

Figure 2: the symbols representing the occurrences of TC are not visible enough on the figures. I would add panels illustrating the temporal evolution of the total accumulated rain and of the TC-induced rain in the three regions. In the legend, you precise that “2 extremely severe cyclones in late October to Early November triggered the largest freshening in the AS”. Please clarify and provide the references for the name (identifier) of these 2 storms and show the SSS freshening associated to them or do not mention something you did not demonstrate. According to Fig2a in the MS, the freshening after September- December 2020, or 2021, seems the two largest freshenings over the period. Satellite SSS data for the example of TC MAHA shown above and which occurred in early November 2019 do not reveal large surface freshening!

Thank you for pointing this out. While the cyclone symbols in Figure 3c are less visible against the blue background, the same symbols used in the other two panels can be referenced for consistency.

The salinity observations presented in Figure 3 aim to demonstrate the overall freshening observed across the northern, central, and southern Arabian Sea. To further clarify the drivers of

this freshening, the separate contributions of cumulative precipitation anomalies, as shown in Figure 2c, provide additional insights into the sources of salinity changes resulting from 2019 tropical cyclones. Additionally, new Figure 9 quantify the impact of cyclone-induced precipitation on ASHSW freshening in 2019. Please note that we have repositioned Figure 3 and the associated discussion earlier in the manuscript, following Figure 1.

We did not intend to suggest that the salinity freshening observed was the largest on record. Rather, our intention was to emphasize that the tropical cyclones in 2019 initiated a salinity freshening event, which persisted through 2023. We have modified the sentence in the revised manuscript for greater clarity.

Figure 3: it is not fully clear how SSS and P anomalies were evaluated. In the method section it is said (Line 476 and 498-499) that ‘salinity and P anomalies are calculated from 2000-2023 period’. Do you mean that you evaluated the mean seasonal cycle over 2000-2023 and then removed it to the S and P time series or did you remove the overall local mean evaluated over the complete 2000-2023 period independently of the season ? Please precise, it is very important to understand if the mean seasonal cycle was removed or not. Please also provide y-axis labels in Fig 3a (SSSA) and b (Precipitation anomaly). Correct the typo for “IMERGE”, it shall reads” IMERG.”

Thank you very much for the feedback. As per the reviewer’s comments, we have clarified the methodology used to calculate the salinity and precipitation anomalies in the revised text. Specifically, we computed a monthly mean climatology (January to December) over the period 2000–2023 and subtracted it from the salinity (average over upper 0–50 m, not SSS) and precipitation time series. This approach effectively removes the long-term monthly cycle associated with monsoon precipitation. This has been clarified in the revised manuscript.

Since Figures 2a and 2b are time-latitude plots, labelling the y-axis with the variable name could introduce confusion. Therefore, we have chosen to omit them for clarity. As suggested, we have also corrected the typo ‘IMERGE’ to ‘IMERG’ Thank you for pointing this error.

Figure 3c: I would suggest to add the TC-induced rain accumulated precipitation time series as well, this would avoid lack of clarity on the relative contributions of TC versus other sources of atmospheric rain on the observed anomalies.

This is an excellent suggestion by the reviewer. By separating the rainfall contributions from tropical cyclones (TCs) and other extreme precipitation events, particularly those in 2020 and 2022, we were able to quantify their relative contributions to the total rainfall more effectively. This analysis provided additional clarity on the individual impacts of TCs versus other sources of rainfall on salinity variability.

Furthermore, the additional model experiments presented in the revised Figure 9 now isolate the rainfall from each source (TCs and extreme precipitation events) and clearly delineate their respective impacts on ASHSW freshening.

As suggested by the reviewer, the relative contributions of rainfall from TCs and extreme events to the total rainfall have also been added in Figure 2c. It is evident from this analysis that rainfall from these events accounted for a significant portion of the excess rainfall observed during the 2019–2023 period. The corresponding model experiments, which incorporated these rainfall sources, effectively explain the ASHSW freshening during this period, further confirming the initial results presented in the manuscript.

We acknowledge that the extreme precipitation event in 2020 was not identified in the earlier version of the manuscript, as it coincided with two cyclones. By applying the 95th percentile method, we have systematically identified and separated their contributions from the total in the revised analysis.

L129: except in 2021, it does not appear clearly that SSS freshening systematically lag behind P anomalies when comparing Fig3a and Fig3b. For example in J-S 2022, the freshening are concomitant with high rain. Please re-formulate accordingly.

As the reviewer points out a clear lag cannot be inferred from Figures 2a and 2b, therefore the sentence is removed. We intend to suggest that the rate of salinity freshening lags behind the precipitation during the 2019–2023 period.

L130: here again, the author sentence “The addition of cyclone-induced freshwater increases buoyancy and strengthens stratification, which in turns weakens the depth of winter convective mixing” has not been yet demonstrated at this stage of the MS and shall be removed if not fully proofed

To address the reviewer’s concern, the sentence has been revised for clarity. The updated sentence now reads: “The addition of freshwater increases buoyancy and strengthens stratification, which in turn reduces the depth of winter convective mixing.”

L134: Fig S2 show the evolution of the MLD averaged over the box 60°-70°E, 15° - 25°N and not separately for the 3 boxes shown in Fig3. It is rather confusing then to compare MLD & S evolution in different boxes. Please correct.

The discussion related to Supplementary Figure 2 has been moved to Figure 4b for improved clarity. Given the limited number of observations in the EN4 dataset, a larger box average was used to better represent the mixed layer depth (MLD) in the ASHSW formation region. Figure 4b shows the MLD from the Argo observations, while Supplementary Figure 2 with the larger box in the northern Arabian Sea broadly capturing the convective formation region of ASHSW.

L135: “While cyclone-induced precipitation weakens the convective formation of ASHSW in the north during winter,...”: here again this statement have NOT been demonstrated at this stage in the MS; please remove/correct unless you show a real demonstration

To address the reviewer’s concern, the sentence has been revised for clarity. The updated sentence now reads: "While precipitation weakens the convective formation of ASHSW in the northern Arabian Sea during winter, ..."

L148: “..salinity maximum at 80 m.” Please precise that is seen in ‘Figure 2f’

Clarified as suggested, the subsurface salinity maximum in Figure 3f occurs at 75 m.

L148: “This vertical salinity stratification promotes vertical mixing (heavier over lighter water) and plays an important role in maintaining SST in the southeastern Arabian Sea”. Ok, but why not showing the Brunt-Väisälä frequency to illustrate the respective contribution of S and T to vertical stratification in the 3 regions, following Maes, C., and O’Kane (2014):

$$N^2(T, S) = -g/\rho \partial \rho / \partial z \approx (g\alpha \partial T \partial z + g\beta \partial S \partial z) = NT^2(T, S) + NS^2(T, S)$$

Where ρ is density, T temperature; S salinity, α is the thermal expansion coefficient, β is the haline contraction coefficient; g is acceleration of gravity and z is vertical. This would help the reader to understand the role of freshwater on the vertical density stratification.

Thank you again for the detailed description. Salinity stratification has been identified as a significant process regulating SST in the southeastern Arabian Sea, with implications for cyclone formation and the onset of the Indian Summer Monsoon, as documented in multiple studies, including Reference (20). In our study, we suggest that recent salinity freshening in the eastern Arabian Sea could further exacerbate SST warming.

The primary focus of this paper is to investigate how freshwater input in the northern Arabian Sea, the formation region of ASHSW, influences stratification and the convective formation of ASHSW. To address this, we have already presented stratification for the northern region using Argo observations in Figure 4c.

However, for clarification, we computed separate contributions of temperature and salinity to the stratification following the methodology suggested by the reviewer and they are shown in **Figure R1** for (a) north and (b) southern regions of the Arabian Sea.

Figure R1. Stratification control. To quantify the relative contributions of temperature and salinity to stratification, the density ratio, R_ρ (Ruddick, B. 1983. *A practical indicator of the stability of the water column to double diffusive activity*. *Deep-Sea Res.*, 30A, 1105–1107, [https://doi.org/10.1016/0198-0149\(83\)90063-8](https://doi.org/10.1016/0198-0149(83)90063-8)) is calculated from potential temperature and salinity observations in the (a) northern and (b) southern regions of the Arabian Sea (refer to Supplementary Figure 3 for location) from 2014 to 2023. The density ratio is defined as: $R_\rho = -\frac{N_\theta^2}{N_s^2}$, where the total stratification (N^2) is expressed as: $N^2 = -\frac{g}{\rho_0} \frac{\partial \rho_\theta}{\partial z} = N_\theta^2 + N_s^2$, with temperature stratification (N_θ^2) and salinity stratification (N_s^2) is defined as: $N_\theta^2 = g\alpha \frac{\partial \theta}{\partial z}$, $N_s^2 = -g\beta \frac{\partial s}{\partial z}$. Here, g is the gravitational acceleration, α is the thermal expansion coefficient of seawater, and β is the saline contraction coefficient of seawater. For $R_\rho \geq 1$, ocean stratified by temperature, $R_\rho \leq -1$, ocean stratified by salinity, and $-1 < R_\rho < 1$, ocean stratified by both temperature and salinity. For clarity, regions where $R_\rho \leq 1$ (i.e., salinity stratification is significant or dominant) are shaded in the plots. Although the water column is predominantly stratified by temperature (white regions), salinity contribution becomes particularly important during winter in the northern Arabian Sea. This salinity influence is in the upper 100 m due to ASHSW and below 150 m due to the intrusion of Persian Gulf Water. In the southern Arabian Sea, salinity stratification arises from the relatively low-salinity water from the Bay of Bengal and the advection of ASHSW from the northern Arabian Sea. The closer R_ρ is to -1, the stronger the opposing influence of salinity on stratification compared to temperature.

L168-170: Why does the extreme precipitation during the summer monsoon of 2017 in the northeastern Arabian Sea reported in reference 7 do not appear in the anomalies of Figure 3b ?

The extreme precipitation event during 2017 was highly localized around 67°E, 22°N. We presume that the use of a larger area average (60°–75°E) may have weakened the amplitude of this event. However, despite its weaker signal, the event is still evident in Figure 2b as a positive precipitation anomaly north of 20°N. In contrast, the extreme precipitation events identified in 2020 and 2022 show pronounced precipitation anomalies (Figures 2b and 2c).

L192 and Figure 4: 1% change in density and 10m shoaling of the MLD are small changes. Can you provide error bars on the Fig4 curves given the rather small number of Argo float profiles you used to estimate these time series (4 as described in the Method section) ? Would be nice to split the contributions of S & T to the PEA following the above split for N2.

As suggested by the reviewer, error bars have been included in Figure 4. The differences in the mean between the two periods are statistically significant at the 95% confidence level ($p=0.05$). We also acknowledge that the mean difference in MLD is specific to the winter period (December–February) and the difference between the two periods is ~20 m (not 10 m as previously noted).

Regarding the partitioning of PEA into relative contributions from temperature and salinity, we are not aware of a standard methodology for directly separating these effects. However, we included separate contributions of stratification from temperature and salinity in Figure R1.

L205-206: “..coinciding with recent freshening events associated with extreme cyclones”: this has not been fully proven and is a gain a fast conclusion. I agree that the stronger PEA over 20219-2023 is coinciding with more cyclones over that period, but has discussed before, these are not necessarily and systematically associated with freshening events: so it is not a proof of causality The 2019-23 anomalies of PEA around the mean do not systematically coincide with TC occurrences. For example, end 2020, 2021 and 2023 the last TCs of the season coincide with a locally decreasing PEA.

To address the reviewer's concern, we have modified the sentence to refer to "extreme precipitation events" instead of "cyclone-induced precipitation," reflecting the separation of these events as discussed in Figure 2c.

L209. Unclear meaning. What do you mean by “similarity”? Can you provide the correlation coefficient between both quantities?

Thank you for pointing this out. The sentence has been omitted in the revised text. Our original intent was to highlight the common trend between the weakening of stratification (indicated by larger PEA) and the depth of winter convective mixing.

L217-219. You state “In 2018, despite the occurrence of three cyclones, none induced freshening north of 15°N, where ASHSW formation takes place. However, in 2019, four out of five cyclones penetrated north of 15°N, resulting in the largest salinity anomalies.” It would be more

correct to say that the increase number of TC in 2019 is “coincident” with the largest salinity anomalies: at this step in the MS, you have not clearly proven that the large SSS drop observed is dominantly due to TC-induced freshening’s.

With the additional analyses that separate the tropical cyclone rainfall contribution from the total rainfall, along with the supporting model experiments, we believe it is now evident that TC activity was responsible for the largest salinity anomaly observed in 2019. Would it not be appropriate to make qualitative inferences, particularly when those inferences are later validated and supported by quantitative results presented in the manuscript?

L220: by the “contrasting salinity patterns observed in the Argo data between the 2018 – 19 and 2019 – 20 periods”:, do you mean the step decrease in the average SSS after mid 2019 as observed in Fig S4, top panel ? please precise.

This section has been revised for clarity as suggested. The sentence now specifically refers to the sharp decrease in salinity observed after mid-2019, in contrast to the higher salinity levels during the 2018–2019 period.

Figure 5: would be nice to add in the time series panel the one of the TC-induced accumulated rain and of the total rain. While indicative, the symbols showing when TC occurred and their increased number is not a direct proof of their impacts on the reported decreased volume of high SSS waters.

Following the reviewer's instructions, TC-induced rainfall has been separated from the total rainfall in Figure 2c and reiterated in Figure 5e. The influence of precipitation on ASHSW volume can be both immediate and delayed, as precipitation affects ASHSW formation locally but may have a delayed impact in the following summer due to southward advection. The perturbation experiments PPE_{2019–20} and PPE_{2018–19} further isolate the specific role of precipitation in modulating ASHSW volume.

Line 240-241. You stated: “The volume of high salinity water revealed a notable decline following each cyclone-induced precipitation event (Figure 5e)”. This is not true in 2021 where the first cyclone coincides with an increase in the high salinity water volume, nor for the second of the year, which was followed by a relatively stable high salinity volume (Figure 5e). Please correct accordingly.

Thank you for the suggestion. This has been corrected accordingly. The statement referred specifically to the 2019 period. While it is true that the volume of ASHSW shows a slight increase following the first storm in June 2021, it subsequently decreases during the winter of 2021–2022, with further reduction by February 2022, influenced by the second storm in September 2021. A reduction in ASHSW volume is expected during the formation period of ASHSW and not immediately following a storm in summer. The impact of rainfall in summer 2021 on the reduction in ASHSW volume becomes evident in the following winter.

Line 312-313: “With the exception of 2022, the agreement between cyclone induced precipitation and the intensification of salinity freshening is clear.” ...and of 2021”. You have not

provided any measure of the “cyclone-induced precipitation” with respect total precipitation, so this is a fast conclusion. Please rephrase.

By separating the rainfall contributions from tropical cyclones (TCs) and other extreme precipitation events, specifically those occurring in 2020 and 2022, we are able to quantify the relative contributions of each to the total rainfall. This analysis provided additional clarity on the roles of TCs versus other rainfall sources in the observed precipitation patterns. Furthermore, the additional model experiments presented in the newly added Figure 9, which isolate rainfall from each source (TCs and extreme precipitation events), clearly delineate their respective impacts on ASHSW freshening.

L323-325: “Thus this experiments provide additional confirmation.....are largely attributable to freshwater input from tropical cyclone”. No, they provide additional confirmation that the change in Precipitation forcing occurring after 2019 is responsible for the decreasing SSS and subsequent weakening of the convective process. Unless, you extract the relative contribution of TCs to total P, you can not state that firmly.

To address the reviewer’s concerns, we provided additional analyses and model experiments. By separating the rainfall contributions from tropical cyclones (TCs) and other extreme precipitation events—specifically those occurring in 2020 and 2022—we are able to quantify the relative contributions of each to the total rainfall. This analysis provided greater clarity on the roles of TCs versus other sources of rainfall. Furthermore, the additional model experiments presented in the newly added Figure 9, which isolate rainfall from each source (TCs and extreme precipitation events), clearly delineate their respective impacts on ASHSW freshening.

The relative contributions of rainfall from TCs and extreme precipitation events to the total rainfall have been added to Figure 2c, as suggested by the reviewer. It is evident that rainfall from these events accounted for a significant portion of the excess rainfall observed during the 2019–2023 period. The model experiments further confirm that these events played a key role in the observed ASHSW freshening during this time, supporting the conclusions drawn earlier in the manuscript.

The intent of this experiment is to capture the salinity differences between the periods 2014–2018 and 2019–2023. The choice of 2018 as a reference year was based on the fact that precipitation anomalies during 2018 are broadly representative of typical conditions observed between 2014 and 2018 (Figure 4). Additionally, the tropical cyclone activity in 2018 reflects typical cyclone tracks commonly observed during this period.

L330-331: can you provide an estimate of the river runoff impacts (see previous comments)

While rainfall from tropical cyclones (TCs) and extreme precipitation events could potentially increase river runoff, particularly as some TCs made landfall along the west coast of India, likely contributing to elevated runoff, the model simulations presented in this study do not account for such effects. The simulations utilized a monthly climatology derived from data prior to 1990, which excludes the impact of river runoff associated with the recent increase in TC activity.

Therefore, the salinity freshening depicted in the model results reflects precipitation-driven effects alone, without contributions from river runoff linked to TCs and extreme events. Although the influence of river runoff is an important topic worth further investigation, it falls beyond the scope of the present study. A brief description of river runoff treatment has been added to the Methods section for clarity.

L349-350: it is not true each year: in 2021, the salinity tendency difference do not exceed 0.1×10^{-3} psu.d⁻¹. Please correct

Corrected as suggested by the reviewer. As noted, the salinity tendency term in 2021 is approximately 0.05. The revised sentence now reads: "Each year, the salinity tendency term exhibits significant decreases, with values ranging from approximately 0.05 to 0.15×10^{-3} psu d⁻¹."

L352-353. In line 336, you claim that due to decrease S with depth in the AS, vertical mixing consistently brings lighter water from below into the mixed layer, so it is confusing to read here that the vertical mixing tends to increase salinity. Please clarify. Is it the difference of vertical mixing between the control and perturbation experiment which is responsible for the increase S (blue curve in Fig 8c)?

Thank you for pointing this out. This point has been clarified in the revised text. The vertical mixing generally reduces salinity, as shown in Figures 8a and 8b. Figure 8c presents the difference in vertical mixing between the control and perturbation experiments. In this case, the reduction in salinity due to vertical mixing in the control experiment is smaller than in the perturbation experiment, resulting in a net increase in salinity in the difference plot.

L357-358: in 2021, the pre- and post moonson TC do appear to have extended the duration of freshwater input but not their amplitude.

The reviewer is correct. The sentence was intended to convey the role of tropical cyclones in prolonging the freshwater input. This has been clarified in the revised text for accuracy.

L361-368: since the beginning of the MS, you keep claiming that TCs are responsible for the increased P and surface freshening but you explain here that in 2022, there were no TCs and that low pressure system can contribute significantly to the observed freshening. This is very confusing and lead the reader to believe that potentially, other low-pressure systems than TCs can be responsible for the observed freshwater fluxes in other years of 2019-23. Please clarify by extracting the actual TC-contribution to rainfall.

Thank you for bringing this up. It is fair to say that the potential contribution of the 2022 event to salinity freshening was mentioned earlier in the manuscript before reaching the conclusions presented here. Moreover, the significant influence of the 2022 extreme rainfall event on ASHSW freshening became increasingly evident in the salt-budget analyses. This extreme event has also been well documented in a series of recent studies.

Although the primary focus of the paper is the dominant influence of rainfall from tropical cyclones in the region north of 15°N, the role of prolonged extreme rainfall events is equally important. Our analysis demonstrated that the extreme events in 2020 and 2022 played a crucial role in sustaining ASHSW freshening. Since such events are rare but appear to be increasing in frequency in recent years, they are distinct from typical Indian summer monsoon rainfall anomalies and instead result from prolonged large-scale organized weather systems like monsoonal depressions. Therefore, they can be characterized as extreme precipitation events similar in magnitude to rainfall from named tropical cyclones.

In the revised manuscript, we have clearly separated the precipitation from tropical cyclones and extreme precipitation events using both observations and model experiments designed to quantify the contribution of each source to salinity freshening. To improve clarity, we now explicitly distinguish these precipitation sources early in the paper (Figure 2c) so that the results presented in the remainder of the manuscript can be interpreted accordingly. This distinction is summarized in paragraph 2, lines xx–yy.

References

- Behara, A., Vinayachandran, P. N., & Shankar, D. (2019). Influence of rainfall over eastern Arabian Sea on its salinity. *Journal of Geophysical Research: Oceans*, 124, 5003–5020. <https://doi.org/10.1029/2019JC014999>
- Boutin, J., N. Reul, J. Koehler, A. Martin, R. Catany, S. Guimbard, F. Rouffi, et al.. Satellite-Based Sea Surface Salinity Designed for Ocean and Climate Studies. *Journal of Geophysical Research*. 2021. <https://doi.org/10.1029/2021JC017676>
- Chaudhuri, D., Sengupta, D., D'Asaro, E., Venkatesan, R., & Ravichandran, M. (2019). Response of the salinity-stratified Bay of Bengal to cyclone Phailin. *Journal of Physical Oceanography*, 49, 1121–1140. <https://doi.org/10.1175/JPO-D-18-0051.1>
- Camberlin, P., Assowe Dabar, O., Pohl, B., Mohamed Waberi, M., Hoarau, K., & Planchon, O. (2024). Contribution of western Arabian Sea tropical cyclones to rainfall in the Horn of Africa and southern Arabian Peninsula. *Journal of Geophysical Research: Atmospheres*, 129, e2024JD041109. <https://doi.org/10.1029/2024JD041109>
- Chen, F. and Y. Fu, (2015) Contribution of tropical cyclone rainfall at categories to total precipitation over the Western North Pacific from 1998 to 2007, *Sci. China Earth Sci.* 58: 2015. doi:10.1007/s11430-015-5103-9.
- Grodsky, S. A., Reul, N., Lagerloef, G., Reverdin, G., Carton, J. A., Chapron, B., et al. (2012). Haline hurricane wake in the Amazon/Orinoco plume: AQUARIUS/SACD and SMOS observations. *Geophysical Research Letters*, 39, L20603. <https://doi.org/10.1029/2012GL053335>
- Jiang, H. and E.J. Zipser, (2010) Contribution of Tropical Cyclones to the Global Precipitation from Eight Seasons of TRMM Data: Regional, Seasonal, and Interannual Variations. *Journal of Climate* 23:6, 1526-1543.

- Jiang, H., C. Liu, and E. D. Zipser (2011), A TRMM-based tropical cyclones cloud and precipitation feature database, *J. Appl. Meteorol. Climatol.*, 50, 1255–1274.
- Li, B., Zhou, L., Qin, J., & Murtugudde, R. (2022). Increase in intraseasonal rainfall driven by the Arabian Sea warming in recent decades. *Geophysical Research Letters*, 49, e2022GL100536. <https://doi.org/10.1029/2022GL100536>
- Lonfat, M., F. D. Marks, and S. S. Chen, (2004) Precipitation distribution in tropical cyclones using the Tropical Rainfall Measuring Mission (TRMM) microwave imager: A global perspective. *Mon. Wea. Rev.*, 132, 1645–1660, doi:10.1175/1520-0493(2004)132,1645:PDITCU.2.0.CO;2.
- Maes, C., and T. J. O'Kane (2014), Seasonal variations of the upper ocean salinity stratification in the Tropics, *J. Geophys. Res. Oceans*, 119, 1706–1722, doi:10.1002/2013JC009366.
- Melnichenko, O., 2021. Multi-mission L4 Optimally Interpoated Sea Surface Salinity. Ver.1.0. PO.DAAC, CA, USA. Dataset accessed [YYYY-MM-DD] at <https://doi.org/10.5067/SMP10-4U7CS>
- Neethu, C. (2018). Insights into the haline variability induced by cyclone Vardah in the Bay of Bengal using SMAP salinity observations. *Remote Sensing Letters*, 9(12), 1205–1213. <https://doi.org/10.1080/2150704X.2018.1519271>
- Rao, A. D., M. Joshi, and M. Ravichandran (2009), Observed low-salinity plume off Gulf of Khambhat, India, during post-monsoon period, *Geophys. Res. Lett.*, 36, L03605, doi:10.1029/2008GL036091.
- Reul, N., Quilfen, Y., Chapron, B., Fournier, S., Kudryavtsev, V., & Sabia, R. (2014). Multisensor observations of the Amazon-Orinoco river plume interactions with hurricanes. *Journal of Geophysical Research: Oceans*, 119, 8271–8295. <https://doi.org/10.1002/2014JC010107>
- Reul Nicolas, Chapron Bertrand, Grodsky Semyon A., Guimbard Sebastien, Kudryavtsev Vladimir, Foltz Gregory R., Balaguru Karthik (2021). Satellite observations of the sea surface salinity response to tropical cyclones . *Geophysical Research Letters* , 48(1), e2020GL091478 (10p) .
- Reul N, Ifremer / LOPS. 2023. Atlas of Tropical Cyclone Induced Wakes (2010-2020) (v1.0) for ESA Marine Atmosphere eXtreme Satellite Synergy project (MAXSS). Ver. 1.0. Ifremer, Plouzane, France. Dataset accessed [2024-10-11].
- Sun, J.; Vecchi, G.; Soden, B. Sea Surface Salinity Response to Tropical Cyclones Based on Satellite Observations. *Remote Sens.* 2021, 13, 420. <https://doi.org/10.3390/rs13030420>
- Yue, X., Zhang, B., Liu, G., Li, X., Zhang, H., & He, Y. (2018). Upper ocean response to typhoon Kalmaegi and Sarika in the South China sea from Multiple-satellite observations and numerical simulations. *Remote Sensing*, 10, 348. <https://doi.org/10.3390/rs10020348>

The relevant references have been included in the revised manuscript to support the points discussed.

Reviewer #3 (Remarks to the Author):

The manuscript “Sustained Freshening of Arabian Sea High Salinity Water Induced by Extreme Tropical Cyclone Precipitation” was reviewed. The paper seems to address an interesting and important issue in these less studied and important ocean basin. It also appears to entail a substantial effort. Hence it can deserve being published. However, some comments listed below may improve it:

We sincerely thank the reviewer for their thoughtful comments and constructive suggestions, which have greatly enhanced the quality of the manuscript and strengthened the scientific arguments presented. We have carefully addressed the majority of the reviewer’s comments and made the necessary modifications to the manuscript. Below, we provide a point-by-point response (reviewer comments in black and our responses highlighted in blue). We hope that these revisions adequately address the reviewer’s concerns.

- 1- The text is well written, but repetitions of some text parts may be reduced (changed) as they are noticeable and can be a bit tedious to the readers? As: “cyclone-induced”, “formation of ASHSW” or “ASHSW formation”, “salinity freshening”. If it can be helped?
- 2- Moving Figs to be more harmonious with text may be useful!?

Thank you for the suggestion. In the revised manuscript, we have minimized repetitive text and ensured consistent terminology throughout the paper. Additionally, we have reorganized the figures to align more closely with the corresponding discussions, as recommended.

- 3- The present work only presents salinity and precipitation records and distributions, and no records of surface heat (especially temperature records from observations or model) and momentum (currents?) fluxes are mentioned (even from the work of the others?) as they are important in understanding the issue being addressed here?

We thank the reviewer for highlighting this important aspect of the study, which was not fully clarified in the initial manuscript. Temperature was excluded in the earlier version because its impact was deemed minimal in the context of our findings. The increasing salinity stratification could indeed restrict vertical mixing by generating barrier layers and potentially raise sea surface temperature (SST) in the northern Arabian Sea. The time-series of temperature observations from Argo floats (Supplementary Figure 4) show no significant differences between the 2014–2018 and 2019–2023 periods. There is, however, a slight indication of warmer winter mixed layer temperatures during the latter period (Supplementary Figure 4a), which could be linked to enhanced salinity stratification from freshwater input. We also added model temperature, which shows good agreement with Argo data, yielding high correlations of 0.84 and 0.91 (Supplementary Figure 8).

Supplementary Figure 4. Argo temperature observations in the Arabian Sea. Along-track Argo float temperature ($^{\circ}\text{C}$) observations for three regions representing (a) northern, (b) central, and (c) southern Arabian Sea (refer to Supplementary Figure 3 for location) from 2014 to 2023. The 25°C contour is overlaid in all panels. Profiles with the presence of a barrier layer is marked with white boxes. Barrier layer thickness (BLT, m) is calculated as the difference between the isothermal layer depth (ILD) and the mixed layer depth (MLD), where MLD is defined as the depth corresponding to a density change associated with a 0.2°C temperature difference from the surface, and ILD defines as the depth where the temperature decreases by 0.2°C from the surface. Profiles with $\text{BLT} \geq 5$ m are marked with white boxes. Cyclone occurrences are marked in each panel: circles indicate very severe to super cyclones, and diamonds indicate cyclones to severe cyclones. Cyclones penetrating north of 15°N are marked in red.

To further investigate the impact of freshwater input on SST, we compared the control (CNTL) and PPE₂₀₁₉₋₂₃ model simulations (Figure R1). The differences in surface heat flux (W m^{-2} , Figure R1a) and SST ($^{\circ}\text{C}$, Figure R1b) between these simulations are negligibly small. A region of net heat gain ($\sim 8 \text{ W m}^{-2}$) is evident along the west coast of India in the control simulation, but the SST exhibited a slight cooling ($\sim 0.1^{\circ}\text{C}$), suggesting the atmosphere tends to warm the ocean through heat input. Due to the negligible magnitude of heat flux, it has not been included in the manuscript.

Figure R1. Differences between the control (CNTL) and PPE₂₀₁₉₋₂₃ experiments (CNTL – PPE₂₀₁₉₋₂₃) for the 5-year period spanning 2019–2023 in (a) surface heat flux (W m^{-2}) and (b) sea surface temperature ($^{\circ}\text{C}$).

While we do not present direct ocean current data, their influence on salinity freshening is captured in the salinity budget analysis (Figure 8). Horizontal and vertical advection terms quantify the salinity changes induced by ocean currents, which remain small in the northern Arabian Sea during winter. For a more comprehensive analysis of heat fluxes and advective processes, we reference Thoppil et al. (2022), Prasad and Ikeda (2002b) for detailed examination of these mechanisms.

4- No cyclone precipitation in 2022 but systemic (synoptic, as large size cyclone or depression) precipitation is substantial!, but the cyclone-induced precipitation is often emphasized? Systemic precipitation have also contributed in other years and cyclone-induced ones are also short lived! Shifting pattern of precipitation northwards on the whole may be more important? Therefore, the paper title may be changed?

We greatly appreciate the reviewer’s insightful suggestion. As noted, large-scale organized weather systems like monsoonal depressions can produce substantial precipitation comparable to that of a significant tropical cyclone. Despite not being associated with a named cyclone, the precipitation from these systems had a comparable effect on salinity freshening (Figure 8). While the extreme precipitation event in 2022 was widely documented, we identified a similar event in 2020, although it coexisted with two tropical cyclones.

In the revised manuscript, we have:

1. Separated precipitation contributions from tropical cyclones and extreme precipitation events using IMERG observations (Figure 2c).

2. Conducted three additional model experiments to quantify the individual and combined effects of tropical cyclones and extreme precipitation events (2020 and 2022) on salinity freshening during 2019–2023 (new Figure 9).

These extreme precipitation events share a key characteristic: they all occurred in the northern Arabian Sea, where ASHSW forms and where extreme freshwater input events have become increasingly frequent in recent years. This trend coincides with a northward shift in tropical cyclone activity, further intensifying freshwater input and salinity freshening. Additionally, the increasing number of extreme post-monsoon tropical cyclones, which are projected to rise with ongoing climate warming, suggests a clear northward shift in their occurrence (Murakami et al., 2017).

Since the primary focus of this study is the freshening of ASHSW driven by freshwater input, and no prior studies have indicated a systematic northward shift in precipitation patterns over the Arabian Sea, emphasizing this aspect in the title may be difficult to justify. However, to better reflect the range of precipitation sources analyzed in this study, we have revised the title to: "*Sustained Freshening of Arabian Sea High Salinity Water Induced by Extreme Precipitation Events.*" The use of "extreme precipitation events" accounts for both tropical cyclone-induced rainfall and large-scale organized weather systems contributing to the observed salinity freshening.

ITCZ in summer, appear to have shifted more (it usually change longitudinal position much more in Indian ocean than in other equatorial regions during the year, hence monsoonal systems) northwards in recent years in the Indian Ocean region that may be due to recent fast climate change, this could have increased the overall precipitation in the Arabian sea?[Mamalakis, A., Randerson, J.T., Yu, J.Y. et al. Zonally contrasting shifts of the tropical rain belt in response to climate change. *Nat. Clim. Chang.* 11, 143–151 (2021).]

Thank you for directing us to this insightful study. While the results indeed project a northward shift of the ITCZ and associated precipitation over the Arabian Sea based on climate model simulations for the period 2075–2100, directly linking this projection to the recent increase in precipitation without a detailed analysis using contemporary data remains challenging. The timescales and uncertainties involved necessitate further investigation using recent observations to establish a clearer connection between observed precipitation patterns and projected long-term trends. In the revised manuscript, we have acknowledged this study as a possible future scenario that could influence the salinity budget in the Indian Ocean (Discussion, paragraph 5), while emphasizing the need for further research using recent data to draw more definitive conclusions.

5- In Fig 3 they may present temperature records along with shown salinity anomaly, and temperature profile in Fig.2, to show better the density stratification which is claimed to have been “substantially weakened”? What is the cooling effects of cloudiness during precipitation? Temperature records may show this. Although in Fig. 4 potential density shows a decrease, along with the shallowing mixed layer depth. (E.g. paper by Shee et al. Recent changes in the upper oceanic water masses over the Indian Ocean using Argo data, *Sci. Rep.* 2023, 13, 20252, also

show some temperature records that may be worth looking at?). So presentation of temperature profiles and series changes may be recommended.

Thank you very much for the suggested recommendation. Unlike salinity, temperature anomalies (**Supplementary Figure 5**) do not exhibit a distinct cooling/warming pattern during 2019–2023 period compared to 2014–2018. They indicate an overall warming trend consistent with the findings of Shee et al. (2023). However, their analysis, focused on the upper 700 m and excluding data after 2019, limits direct comparison with our results. Notably, there are indications of warming in the northern Arabian Sea in 2020, 2021, and 2022, potentially linked to salinity-driven stratification, which may have restricted vertical mixing.

Related revisions can be found under subsection: Prolonged salinity freshening: Observations, paragraphs, 9 and 11, Figure 4b and supplementary Figures 4, 5 and 8.

Supplementary Figure 5. Temperature anomalies ($^{\circ}\text{C}$) averaged in the upper 50 m from the monthly EN4 dataset. Anomalies are calculated relative to the 2000–2023 period by subtracting the long-term monthly climatology.

During tropical cyclone periods, latent heat flux predominantly contributes to surface cooling, though reductions in solar radiation due to cloud cover also modulate surface heat flux to a lesser extent.

As recommended by the reviewer, we have included Argo temperature profiles in Supplementary Figure 4 for clarity, which show similar slightly warmer winter temperature during 2019–2023, likely driven by strong salinity stratification.

6- As the northern Indian Ocean is experiencing the intrusions of high salinity and temperature at depth of about 100 to 500 m from the Persian Gulf and Red Sea outflows (depending on season) while being refreshed by the surface water (especially the Persian Gulf, which is also refreshed by the surface water of Arabian Sea), have these semi-enclosed seas any influence on the ASHSW? It may be worth commenting on this, in discussion?

This is an excellent suggestion by the reviewer to clarify the influence of different water masses on ASHSW. We have added a brief note in the discussion to address this point (see Discussion, paragraph 3). As noted, ASHSW is confined to the upper 100 m, while Persian Gulf Water occurs below 150 m, with its core around 200 m depth, and Red Sea Water is typically found below 500 m, spreading eastward. Therefore, these deeper water masses do not interact with ASHSW. The

model experiments, which consistently represent these water masses, further support this distinction.

7- Arabian Sea oxygen- depleted waters and hence, fish kill has increased recently, can ASHSW changes influence this; this may be interesting to just point out? Oxygenation of the Oman Sea and northern Indian Ocean waters by these outflows (4) has been shown to be important, although they are at deeper depths (e.g. Font, E., Swart, S., Bruss, G., Sheehan, P. M. F., Heywood, K. J., & Queste, B. Y. (2024). “Ventilation of the Arabian Sea oxygen minimum zone by Persian Gulf water”. *Journal of Geophysical Research – Oceans*, 129(5), [e2023JC020668]).

Thank you for pointing out this interesting recent study on the role of Persian Gulf Water in the Arabian Sea Oxygen Minimum Zone (OMZ). It is highly unlikely that ASHSW contributes to the OMZ, which occurs below 150 m, as ASHSW is primarily confined to the upper 100 m of the water column. We are unaware of any evidence linking ASHSW to the OMZ in the Arabian Sea. However, the weakening of winter convective mixing can directly affect primary productivity in the northern Arabian Sea, which may indirectly impact fish populations dependent on phytoplankton. This point has been incorporated into the Discussion (paragraph 4).

8- Fig. 1 show cyclone track shifted northwards, is this due to recent climate change? Not clearly reasoned? Precipitation seems to have increased more over the eastern coast of the Arabian Sea (more summer ITCZ shift, 4, ?).

The reviewer raises an important and relevant point. The observed northward shift in cyclone tracks is consistent with projections under changing climate scenarios. Notably, the recent increase in the frequency of post-monsoon cyclones and their northward migration closely aligns with projected changes in extreme cyclone activity, as shown by Murakami et al. (2017, see their Figure 2, referenced below). This connection was briefly discussed in the earlier version of the manuscript.

Beyond precipitation from cyclones and synoptic-scale systems, Figure 2b also shows an overall increasing trend in precipitation in the eastern Arabian Sea. This rise could be linked to other atmospheric modes of interannual variability, such as the Indian Ocean Dipole (IOD), Monsoon Intra-Seasonal Oscillation (MISO), and monsoon depressions. However, since these processes were not the primary focus of this study, they were only briefly mentioned in the revised manuscript for context.

[REDACTED]

Figure 2 of Murakami, H., Vecchi, G.A. & Underwood, S. Increasing frequency of extremely severe cyclonic storms over the Arabian Sea. *Nature Clim Change* 7, 885–889 (2017).
<https://doi.org/10.1038/s41558-017-0008-6>

9- Should $\Sigma \Delta P$ be + and $\Sigma \Delta S$ be - in Fig. 3(c)? Can these be decadal Oscillation?

Figure 2c (previous Figure 3c) presents the cumulative anomalies of salinity and precipitation depicted in Figures 2a and 2b. During the 2014–2018 period, salinity anomalies remain positive, resulting in a progressively increasing positive cumulative sum, while precipitation anomalies become increasingly negative. In contrast, the 2019–2023 period shows a reversal in the sign of both salinity and precipitation anomalies, leading to a corresponding shift in their cumulative trends.

Determining whether the observed trend is part of a longer-term decadal oscillation would require further analysis using a longer data record. However, the study by Shee et al. (2021), based on 17 years (2003–2019) of Argo data, suggests the presence of decadal oscillations in the

upper 700 m of the northern Arabian Sea, although the study did not explore the underlying processes driving these variations in detail.

10- In salinity budget equation, F_v and F_h are probably vertical and horizontal fluxes due other forcing (may be turbulence or eddy fluxes)? They better to be introduced clearly.

Thank you for the valuable suggestion. As recommended, the details of these terms have been included in the revised manuscript (see Methods) for clarity and completeness.

Best of luck for the author.

Thank you!

Review of the MS entitled “*Sustained Freshening of Arabian Sea High Salinity Water Induced by Extreme Tropical Cyclone Precipitation*” by P.G. Thoppil submitted to Nature Communications.

Summary:

In this MS, the author first analyzes the impact of Tropical Cyclones (TCs)-induced precipitation on upper ocean salinity in the North Arabian Sea (AS) over 2014-2023 using satellite rain and in situ Argo profile data. The author reports an increased number of TC in 2019-2023, concomitant with an increase in total precipitation and an overall decrease of upper ocean salinity in the East AS with respect to the 2014-2018 period, with a maximum freshening observed in the North East AS region. He then analyzes the temporal evolution of the vertical salinity structure using *in situ* profile data, revealing sustained freshening in the Northern Arabian Sea where the Arabian Sea High Salinity Waters (ASHSW) form. The upper ocean freshening reduces the potential surface density, augmenting the vertical stratification and in fine, disrupt the convective sinking process of ASHSW. Finally, using model perturbation experiment and model-based salinity budget estimates, the author attempts to isolate the impact of TC-induced rain on salinity freshening and conclude that the recent SSS freshening is caused by the increased number of TCs from 2019 onward.

Major Comments and recommendations

The topic is of strong scientific interest, in particular for the tropical cyclone forecast, salinity remote sensing, and, ocean modeler communities. While the SST, CHL-A, and SSH response to TC passage have been extensively studied, TC-induced SSS changes and their large-scale impacts on other climate processes are much less known: this work definitively participate to these community efforts. In general, the MS reads well, is interesting and the results are in general plausible. However, I have several major concerns.

First, in the part of the MS devoted to observations analyses, the author had not at all isolated the TC-induced precipitation and associated SSS changes from other sources of atmospheric and oceanic variability, which weaken strongly the strength of his statements, which are often more ‘suggestions’ than a clear demonstration of causality between the three reported interannual anomalies seen in the observation: 1) ‘increased number of TC’, 2) ‘increased total Precipitation’, and 3) ‘decreased overall upper ocean Salinity’. For me, the author has not demonstrated convincing evidences from his analyses that the reported large-scale upper ocean salinity and precipitation anomalies (shown in Figure 1 & Figure 2 in the MS), as well as their consequences on the stability of the water column (Figure 4) is dominantly due to the passage of some rare but intense tropical cyclones rainfall and associated SSS freshening. I will show below in the detailed comments section of my review several examples in 2019 where there was little precipitation accumulated during the passage of some strong TCs in this region as well as small associated SSS changes in their wake, as reported from satellite observations. The

interannual SSS & P changes observed by the author could therefore be due to other processes (see details in Behara et al, 2019) such as the increased precipitation during the recent Monsoon, or e.g., due to increased runoff from the river waters out of the Gulf of Khambhat estuaries, or from elsewhere in the coasts (e.g Pakistan floods in 2022 as mentioned by the author at the end of his MS), or even more likely by large scale climate variability impact on rain and SSS (e.g., Indian Ocean Dipole, Moonsoon IntraSeasonal Oscillation (MISO),...) and not necessarily because of changes in Tropical Cyclone activity *per se* in between 2019-2023 and 2014-2018. For example, Li et al (2022) found that the variance of MISO has an increasing trend from 1982 to 2017 over the northeastern Arabian Sea (AS), accompanied by increasing intraseasonal rainfall. How does this impact the precipitation excess shown in Figure 1c? The enhancement in rainfall in this case is mainly nourished by the increasing moisture supply, with a major contribution from the upward moisture advection by the intraseasonal vertical wind velocity. Since the author did not isolate the precipitation signal from TC-only with respect to total rain, and also did not isolate the salinity response to TC passage, most of his statements remain poorly evidenced, while this can be now easily estimated from the observation dataset already gathered by the author:

- 1) From IMERG and Ibtracs datasets, the author could estimate the rain accumulated during the passage of each storm over 2014-2023 in a spatial domain (e.g. within the wind radius of 34knot winds or within a radius of 500 km) around the TC center tracks.
- 2) Then further estimate the time series of the cumulated contribution of tropical cyclone only induced rainfall to the total precipitation over the 3 AS subregions selected by the author. Then he could compare the relative contribution of TCs separately for both period 2014-2018 and 2019-2023.

The respective contribution of TC-induced precipitation to total rain and their interannual anomalies could then be much better estimated and would strongly re-enforce the author' statements. Such approach has been used frequently in the past literature based on satellite rain products and could be applied following for example the approaches proposed in Lonfat et al (2004), Chen and Fu (2015), Jiang and Zipser (2010), Jiang et al (2011). For example, a recent study by Camberlin et al. (2024) evaluated the contribution of the Western Arabian Sea Tropical Cyclones to Rainfall in the Horn of Africa and Southern Arabian Peninsula by estimating rainfall on non-tropical cyclone (TC) days and on TC days based on IMERG data (2000–2020).

Similarly, to best isolate the TC-induced surface freshening from other sources of SSS variability (e.g., Moonsoon rain, horizontal advection, river runoff impacts), **the SSS change induced by each TC can now be evaluated systematically with confidence using satellite SSS data** back to 2010 following the approaches presented in numerous papers in the literature (see for examples Chaudhuri et al., 2019; Grodsky et al., 2012; Neethu, 2018; Reul et al., 2014; Yue et al., 2018, Reul et al. 2021 and Sun et al., 2021). Merged SMOS, Aquarius and SMAP high quality Level 4 satellite SSS products now

exist to perform such analyses (e.g., ESA Climate Change Initiative SSS products, Boutin et al. 2021 or the Multi-mission L4 Optimally Interpoated Sea Surface Salinity, Melnichenko, 2021). **The author could then evaluate the cumulated TC-induced SSS changes** in each season over 2014-2023 and see how much it contributed to the overall SSS decrease observed in that region since 2019.

Second, in the second part of the MS devoted to model analyses, here again the author has not directly isolated the impact of interannual variability of the TC-induced precipitation and associated SSS changes from other sources of interannual atmospheric and oceanic variability. While I agree with the author that by inverting atmospheric forcing of years 2018 and 2019 the model can reproduce the SSS freshening observation: it does only show that the interannual variability in P predominantly modulate the SSS signal: the author however did not show directly that the TC-induced rain only is responsible for that interannual signal: some other sources of precipitation variability over 2018-2019 could be responsible for it. Once the TC-induced rain will be isolated from the total precipitation time series as I suggested previously, the author could then easily re-run his model perturbation experiments by only removing TC days in the atmospheric forcing time series: this would directly show what the author want to prove rather than by using lengthy and sometimes confusing indirect approaches.

In addition, many time in the MS, the author used very strong statements (particularly on the impacts of TC-precipitation on SSS anomalies) without full proof of them: please rephrase those sentences to only state what is shown by your results.

Therefore, despite most of the results shown by the author are plausible, I recommend that the MS undergo major revision before being considered for publication so that the author actually and properly separate the TC impacts from other sources of variability in both the Precipitation and SSS signals and in the observation and model analyses, separatly: this shall in fine directly demonstrate the authors indirect statements.

Details are given below.

Detailed Comments:

L38 “evaporation-precipitation”- What about river runoff and their impacts on the north-east Arabian Sea (AS) upper salinity, in particular from the Indus and the Narmada, Sabamati, Mahi and Tapi rivers in the Gulf of Khambhat estuaries ? As shown in Rao et al (2009) and reproduced here below (panel d), rivers have non negligible impacts on the SSS in the North eastern AS, particularly in winter months.

[REDACTED]

Figure 1: Monthly climatological salinity for (a) July, (b) October, (c) December and (d) ARGO observations averaged for 17–20N during 2004. From Rao et al (2009)

L107-108: this sentence is a very strong statement that definitively need more support than what is shown and discussed around Fig1 at this stage in the MS. Please re-phrase to be more accurate and keep to the point of what your data really show. It is not because Fig1 panels reveal a concomitant increase in 1) the number of TCs in the North-eastern part of the AS, 2) as well as in overall Precipitation in the eastern AS and 3) a decrease in SSS in the NE AS that this is clearly an evident proof that “the cyclone-induced precipitation have impact on the convective formation of the Arabian Sea High Salinity Waters “. This is a very strong shortcoming. As I will show further, TC in the Arabian sea are not systematically associated with heavy rain and SSS drops are not systematically observed in their trail. Deeper observation analyses are required to conclude.

Indeed, the increase of precipitation shown in Fig1c could be due to other atmospheric sources of interannual variability (e.g., Indian Ocean Dipole, Monsoon) than merely the change in Tropical Cyclone activity in between 2019-2023 and 2014-2018. What about a potentially increase in monsoon season accumulated rain in the most recent years ?

Similarly, the SSS decrease in the AS, while I agree is most intense where most new TCs tracks where located in 2019-2023 is also a region of significant runoff from the Narmada, Sabamati, Mahi and Tapi rivers in the Gulf of Cambay estuaries. Could the reported interannual variability in SSS in this region be also related to increased river runoff in 2019-2023 ? and if yes, how much does it contribute to the observed SSS decrease with respect to the impact of local rain increases ?

To resolve these uncertainties, I suggest the author shall re-analyse the datasets as follows:

- 3) From IMERG and Ibracs datasets, estimate the rain accumulated during the passage of each storm over 2014-2023 in a spatial domain (e.g. within the wind radius of 34knot winds) around the TC center track.
- 4) Then further estimate the average cumulated spatial contribution of all tropical cyclone rainfall to total precipitation over the AS region, separately for both period 2014-2018 and 2019-2023.

The respective contribution of TC-induced precipitation to total rain and their interannual anomalies could then be much better estimated and would strongly re-enforce the author's statements. Such approach have been used frequently in the past literature based on satellite rain products and could be applied following for example the approaches proposed in Lonfat et al (2004), Chen and Fu (2015), Jiang and Zipser(2010), Jiang et al (2011). For example a recent study by Camberlin et al. (2024) evaluated the contribution of the Western Arabian Sea Tropical Cyclones to Rainfall in the Horn of Africa and Southern Arabian Peninsula by estimating rainfall on non-tropical cyclone (TC) days and TC days based on IMERG data (2000–2020). Such an approach could be easily applied by the author.

- 5) Similarly, to best isolate the TC-induced surface freshening from other sources of SSS variability (Moonsoon rain, advection, river runoff impacts), the SSS change induced by each TC can now be evaluated systematically with confidence using satellite SSS data back to 2010 following the approaches presented in numerous papers in the literature (see for examples Chaudhuri et al., 2019; Grodsky et al., 2012; Neethu, 2018; Reul et al., 2014; Yue et al., 2018, Reul et al. 2021 and Sun et al., 2021). Merged SMOS, Aquarius and SMAP high quality Level 4 satellite SSS products now exist to perform such analyses (e.g., ESA Climate Change Initiative SSS products, Boutin et al. 2021 or the Multi-mission L4 Optimally Interpoated Sea Surface Salinity, Melnichenko, 2021)

In fact, databases including TC-induced Precipitation & SSS changes for each individual TC globally over 2010-2020 were already developed in the frame of the ESA Marine Atmosphere eXtreme Satellite Synergy (MAXSS) project (see <https://www.maxss.org/>) and could be used by the author to better demonstrate his statements. This dataset is publicly available (doi:[10.12770/cc0577e4-55d6-4aa9-a938-b4965be121ab](https://doi.org/10.12770/cc0577e4-55d6-4aa9-a938-b4965be121ab), Reul et al., 2023), and provides, for each storm tracks from IBTrACS during the period 2010-2020, three types of storm-related parameters: (1) the pre-storm upper ocean conditions (e.g., SST, SSS, upper ocean vertical density stratification strength, etc..), (2) the atmospheric forcing during the storm passage (maximum wind, accumulated rain and evaporation, wind power index, etc..) during the storm, as well as (3) the oceanic wakes left after the storm for an ensemble of key variables (SST, SSS, sea level height, ocean color parameters). I show here below few examples of estimated SSS anomalies (derived from the ESA CCI SSS products) left after TC passages in the AS as well as the corresponding accumulated precipitation (derived from merged satellite TRMM products) during their lifetime over the AS in 2019 and which is available in the

MAXSS database. As evidenced from satellite data, TC passage in the AS in 2019 are not systematically associated with strong local SSS decreases and heavy precipitation: see e.g. examples of the extremely severe cyclonic storm MAHA in 2019 with small amount of accumulated precipitation associated with negligible SSS changes in the North Eastern AS, or the other case of the super cyclonic storm KYARR with locally very high accumulated rain at 20°N but no significant associated SSS changes observed from satellites. Significant SSS drops were nevertheless observed after the passage of the very severe cyclonic storm VAYU in a region with significant accumulated rain. Therefore, given the high variability of the SSS response to TC passages in this region, and although Sun et al., (2021) have shown (see their Figure 16 top pannel) that on average TC do induce SSS drops in the AS, it remains unclear what the overall contributions of the TC-induced rain and associated freshening are on the 2014-2018 versus 2019-2023 interannual variability reported in Fig 1.c and Fig.1d.

Figure 2: Examples of SSS anomalies left after TC passages in the Arabian Sea in 2019 (left panels) and corresponding accumulated rain (right panels). Data and plots are available at <https://www.maxss.org/>.

Line 111-113: I agree that the addition of freshwater reduces surface density but it has not been demonstrated at this stage of the MS that it disrupts the sinking process and weakens the formation of the ASHSW. Please re-phrase to only state what is shown by your results.

Figure 2: the symbols representing the occurrences of TC are not visible enough on the figures. I would add panels illustrating the temporal evolution of the total accumulated rain and of the TC-induced rain in the three regions. In the legend, you precise that “2 extremely severe cyclones in late October to Early November triggered the largest freshening in the AS”. Please clarify and provide the references for the name (identifier) of these 2 storms and show the SSS freshening associated to them or do not mention something you did not demonstrate. According to Fig2a in the MS, the freshening after September-December 2020, or 2021, seems the two largest freshenings over the period. Satellite SSS data for the example of TC MAHA shown above and which occurred in early November 2019 do not reveal large surface freshening!

Figure 3: it is not fully clear how SSS and P anomalies were evaluated. In the method section it is said (Line 476 and 498-499) that ‘salinity and P anomalies are calculated from 2000-2023 period’. Do you mean that you evaluated the mean seasonal cycle over 2000-2023 and then removed it to the S and P times series or did you remove the overall local mean evaluated over the complete 2000-2023 period independently of the season? Please precise, it is very important to understand if the mean seasonal cycle was removed or not. Please

also provide y-axis labels in Fig 3a (SSSA) and b (Precipitation anomaly). Correct the typo for “IMERGE”, it shall reads” IMERG.”

Figure 3c: I would suggest to add the TC-induced rain accumulated precipitation time series as well, this would avoid lack of clarity on the relative contributions of TC versus other sources of atmospheric rain on the observed anomalies.

L129: except in 2021, it does not appear clearly that SSS freshening systematically lag behind P anomalies when comparing Fig3a and Fig3b. For example in J-S 2022, the freshening are concomitant with high rain. Please re-formulate accordingly.

L130: here again, the author sentence “*The addition of cyclone-induced freshwater increases buoyancy and strengthens stratification, which in turns weakens the depth of winter convective mixing*” has not been yet demonstrated at this stage of the MS and shall be removed if not fully proofed

L134: Fig S2 show the evolution of the MLD averaged over the box 60°-70°E, 15° - 25°N and not separately for the 3 boxes shown in Fig3. It is rather confusing then to compare MLD & S evolution in different boxes. Please correct.

L135: “*While cyclone-induced precipitation weakens the convective formation of ASHSW in the north during winter,...*”: here again this statement have NOT been demonstrated at this stage in the MS; please remove/correct unless you show a real demonstration

L148: “..salinity maximum at 80 m.” Please precise that is seen in ‘Figure 2f’

L148: “This vertical salinity stratification promotes vertical mixing (heavier over lighter water) and plays an important role in maintaining SST in the southeastern Arabian Sea”. Ok, but why not showing the Brunt-Väisälä frequency to illustrate the respective contribution of S and T to vertical stratification in the 3 regions, following Maes, C., and O’Kane (2014):

$$N^2(T, S) = -\frac{g}{\rho} \frac{\partial \rho}{\partial z} \approx \left(g\alpha \frac{\partial T}{\partial z} + g\beta \frac{\partial S}{\partial z} \right) = N_T^2(T, S) + N_S^2(T, S)$$

Where ρ is density, T temperature; S salinity, α is the thermal expansion coefficient, β is the haline contraction coefficient; g is acceleration of gravity and z is vertical. This would help the reader to understand the role of freshwater on the vertical density stratification.

L168-170: Why does the extreme precipitation during the summer monsoon of 2017 in the northeastern Arabian Sea reported in reference 7 do not appear in the anomalies of Figure 3b ?

L192 and Figure 4: 1% change in density and 10m shoaling of the MLD are small changes. Can you provide error bars on the Fig4 curves given the rather small number of Argo float profiles you used to estimate these time series (4 as described in the Method section) ? Would be nice to split the contributions of S & T to the PEA following the above split for N2.

L205-206: “..coinciding with recent freshening events associated with extreme cyclones’: this has not been fully proven and is a gain a fast conclusion. I agree that the stronger PEA over 2021-2023 is coinciding with more cyclones over that period, but has discussed before, these are not necessarily and systematically associated with freshening events: so it is not a proof of causality The 2019-23 anomalies of PEA around the mean do not systematically coincide with TC occurrences. For example, end 2020, 2021 and 2023 the last TCs of the season coincide with a locally decreasing PEA.

L209. Unclear meaning. What do you mean by “similarity”? Can you provide the correlation coefficient between both quantities?

L217-219. You state “In 2018, despite the occurrence of three cyclones, none induced freshening north of 15°N, where ASHSW formation takes place. However, in 2019, four out of five cyclones penetrated north of 15°N, resulting in the largest salinity anomalies.” It would be more correct to say that the increase number of TC in 2019 is “coincident” with the largest salinity anomalies: at this step in the MS, you have not clearly proven that the large SSS drop observed is dominantly due to TC-induced freshening’s.

L220: by the “contrasting salinity patterns observed in the Argo data between the 2018 – 19 and 2019 – 20 periods”; do you mean the step decrease in the average SSS after mid 2019 as observed in Fig S4, top panel ? please precise.

Figure 5: would be nice to add in the time series panel the one of the TC-induced accumulated rain and of the total rain. While indicative, the symbols showing when TC occurred and their increased number is not a direct proof of their impacts on the reported decreased volume of high SSS waters.

Line 240-241. You stated: “*The volume of high salinity water revealed a notable decline following each cyclone-induced precipitation event (Figure 5e)*”. This is not true in 2021 where the first cyclone coincides with an increase in the high salinity water volume, nor for the second of the year, which was followed by a relatively stable high salinity volume (Figure 5e). Please correct accordingly.

Line 312-313: “With the exception of 2022, the agreement between cyclone-induced precipitation and the intensification of salinity freshening is clear.” ...and

of 2021”. You have not provided any measure of the “cyclone-induced precipitation” with respect total precipitation, so this is a fast conclusion. Please rephrase.

L323-325: “Thus this experiments provide additional confirmation.....are largely attributable to freshwater input from tropical cyclone”. No, they provide additional confirmation that the change in Precipitation forcing occurring after 2019 is responsible for the decreasing SSS and subsequent weakening of the convective process. Unless, you extract the relative contribution of TCs to total P, you can not state that firmly.

L330-331: can you provide an estimate of the river runoff impacts (see previous comments)

L349-350: it is not true each year: in 2021, the salinity tendency difference do not exceed $0.1 \cdot 10^{-3} \text{ psu.d}^{-1}$. Please correct

L352-353. In line 336, you claim that due to decrease S with depth in the AS, vertical mixing consistently brings lighter water from below into the mixed layer, so it is confusing to read here that the vertical mixing tends to increase salinity. Please clarify. Is it the difference of vertical mixing between the control and perturbation experiment which is responsible for the increase S (blue curve in Fig 8c)?

L357-358: in 2021, the pre- and post moonson TC do appear to have extended the duration of freshwater input but not their amplitude.

L361-368: since the beginning of the MS, you keep claiming that TCs are responsible for the increased P and surface freshening but you explain here that in 2022, there were no TCs and that low pressure system can contribute significantly to the observed freshening. This is very confusing and lead the reader to believe that potentially, other low-pressure systems than TCs can be responsible for the observed freshwater fluxes in other years of 2019-23. Please clarify by extracting the actual TC-contribution to rainfall.

References

- Behara, A., Vinayachandran, P. N., & Shankar, D. (2019). Influence of rainfall over eastern Arabian Sea on its salinity. *Journal of Geophysical Research: Oceans*, 124, 5003–5020. <https://doi.org/10.1029/2019JC014999>
- Boutin, J., N. Reul, J. Koehler, A. Martin, R. Catany, S. Guimbard, F. Rouffi, et al.. Satellite-Based Sea Surface Salinity Designed for Ocean and Climate Studies. *Journal of Geophysical Research*. 2021. <https://doi.org/10.1029/2021JC017676>

- Chaudhuri, D., Sengupta, D., D'Asaro, E., Venkatesan, R., & Ravichandran, M. (2019). Response of the salinity-stratified Bay of Bengal to cyclone Phailin. *Journal of Physical Oceanography*, **49**, 1121–1140. <https://doi.org/10.1175/JPO-D-18-0051.1>
- Camberlin, P., Assowe Dabar, O., Pohl, B., Mohamed Waberi, M., Hoarau, K., & Planchon, O. (2024). Contribution of western Arabian Sea tropical cyclones to rainfall in the Horn of Africa and southern Arabian Peninsula. *Journal of Geophysical Research: Atmospheres*, **129**, e2024JDO41109. <https://doi.org/10.1029/2024JDO41109>
- Chen, F. and Y. Fu, (2015) Contribution of tropical cyclone rainfall at categories to total precipitation over the Western North Pacific from 1998 to 2007, *Sci. China Earth Sci.* **58**: 2015. doi:10.1007/s11430-015-5103-9.
- Grodsky, S. A., Reul, N., Lagerloef, G., Reverdin, G., Carton, J. A., Chapron, B., et al. (2012). Haline hurricane wake in the Amazon/Orinoco plume: AQUARIUS/SACD and SMOS observations. *Geophysical Research Letters*, **39**, L20603. <https://doi.org/10.1029/2012GL053335>
- Jiang, H. and E.J. Zipser, (2010) Contribution of Tropical Cyclones to the Global Precipitation from Eight Seasons of TRMM Data: Regional, Seasonal, and Interannual Variations. *Journal of Climate* **23**:6, 1526-1543.
- Jiang, H., C. Liu, and E. D. Zipser (2011), A TRMM-based tropical cyclones cloud and precipitation feature database, *J. Appl. Meteorol. Climatol.*, **50**, 1255–1274.
- Li, B., Zhou, L., Qin, J., & Murtugudde, R. (2022). Increase in intraseasonal rainfall driven by the Arabian Sea warming in recent decades. *Geophysical Research Letters*, **49**, e2022GL100536. <https://doi.org/10.1029/2022GL100536>
- Lonfat, M., F. D. Marks, and S. S. Chen, (2004) Precipitation distribution in tropical cyclones using the Tropical Rainfall Measuring Mission (TRMM) microwave imager: A global perspective. *Mon. Wea. Rev.*, **132**, 1645–1660, doi:10.1175/1520-0493(2004)132,1645:PDITCU.2.o.CO;2.
- Maes, C., and T. J. O'Kane (2014), Seasonal variations of the upper ocean salinity stratification in the Tropics, *J. Geophys. Res. Oceans*, **119**, 1706–1722, doi:10.1002/2013JC009366.
- Melnichenko, O., 2021. Multi-mission L4 Optimally Interpoated Sea Surface Salinity. Ver. 1.0. PO.DAAC, CA, USA. Dataset accessed [YYYY-MM-DD] at <https://doi.org/10.5067/SMP10-4U7CS>

- Neethu, C. (2018). Insights into the haline variability induced by cyclone Vardah in the Bay of Bengal using SMAP salinity observations. *Remote Sensing Letters*, 9(12), 1205–1213. <https://doi.org/10.1080/2150704X.2018.1519271>
- Rao, A. D., M. Joshi, and M. Ravichandran (2009), Observed low-salinity plume off Gulf of Khambhat, India, during post-monsoon period, *Geophys. Res. Lett.*, 36, L03605, doi:10.1029/2008GL036091.
- Reul, N., Quilfen, Y., Chapron, B., Fournier, S., Kudryavtsev, V., & Sabia, R. (2014). Multisensor observations of the Amazon-Orinoco river plume interactions with hurricanes. *Journal of Geophysical Research: Oceans*, 119, 8271–8295. <https://doi.org/10.1002/2014JC010107>
- Reul Nicolas, Chapron Bertrand, Grodsky Semyon A., Guimbard Sebastien, Kudryavtsev Vladimir, Foltz Gregory R., Balaguru Karthik (2021). Satellite observations of the sea surface salinity response to tropical cyclones . *Geophysical Research Letters* , 48(1), e2020GL091478 (10p.) .
- Reul N, Ifremer / LOPS. 2023. Atlas of Tropical Cyclone Induced Wakes (2010-2020) (v1.0) for ESA Marine Atmosphere eXtreme Satellite Synergy project (MAXSS). Ver. 1.0. Ifremer, Plouzane, France. Dataset accessed [2024-10-11].
- Sun, J.; Vecchi, G.; Soden, B. Sea Surface Salinity Response to Tropical Cyclones Based on Satellite Observations. *Remote Sens.* 2021, 13, 420. <https://doi.org/10.3390/rs13030420>
- Yue, X., Zhang, B., Liu, G., Li, X., Zhang, H., & He, Y. (2018). Upper ocean response to typhoon Kalmaegi and Sarika in the South China sea from Multiple-satellite observations and numerical simulations. *Remote Sensing*, 10, 348. <https://doi.org/10.3390/rs10020348>